# Structure-based virtual screening, molecular docking, and MD simulation studies: An in-silico approach for identifying potential MBL inhibitors

Emira Noumi[1,2]*, Mejdi Snoussi[1,2], Nouha Bouali[1,2], Mamdouh M. Alshammari[1], Hisham N. Altayb[3,4], Muhammad Afzal[5], Vincenzo De Feo[6]

1 Department of Biology, College of Science, University of Ha'il, Hail, Saudi Arabia, 2 Medical and Diagnostic Research Centre, University of Ha'il, Hail, Saudi Arabia, 3 Department of Biochemistry, Faculty of Science, King Abdulaziz University, Jeddah, Saudi Arabia, 4 Faculty of Medical Laboratory Science, Sudan University of Science and Technology, Khartoum, Sudan, 5 Department of Pharmaceutical Sciences, Batterjee Medical College, Pharmacy Program, Jeddah, Saudi Arabia, 6 Department of Pharmacy, University of Salerno, Salerno, Italy

* eb.noumi@uoh.edu.sa

## Abstract

The global rise of antibiotic-resistant infections has been driven in part by the spread of bacteria producing metallo-β-lactamase (MBL), particularly New Delhi metallo-β-lactamase-1 (NDM-1). Currently, there are no clinically approved inhibitors targeting NDM-1 or other MBLs, highlighting the urgent need for novel therapeutic agents. This study addresses this gap by identifying potential NDM-1 inhibitors through a comprehensive *in silico* workflow. A total of 4,561 natural product compounds were screened using a machine learning (ML)-based quantitative structure–activity relationship (QSAR) model. Molecular docking was then performed to prioritize hits, followed by Tanimoto similarity-based clustering to identify representative compounds. The three most promising compounds identified were **S721-1034**, S904-0022, and **N118-0137**. 300 ns molecular dynamics (MD) simulation was used to examine binding interactions and stability of a control molecule (meropenem (0RV)) and the three selected compounds (**S721-1034**, S904-0022, and **N118-0137**) with the target protein. Among the three compounds evaluated, **S904-0022** demonstrated consistent root mean square deviation (RMSD) values throughout the molecular dynamics (MD) simulation compared to the other two ligands. Additionally, **S904-0022** exhibited considerable affinity with key residues, including Gln123, His250, Trp93, and Val73, indicating robust interactions with NDM-1. The strength of this interaction was further validated by a significantly favorable binding free energy of −35.77 kcal/mol, markedly better than the control compound (−18.90 kcal/mol). The strength of this interaction was further validated by a significantly favorable binding free energy of −35.77 kcal/mol, markedly better than the control compound (−18.90 kcal/mol). The findings of this study provide valuable insights into the molecular interactions and stability of these compounds, which can be used to improve drug development and explore

**Data availability statement:** All data supporting the findings of this study are available within the main article and its Supporting information. Additional data, including docking scores and QSAR outputs, are provided in the Supplementary Files (Excel format).

**Funding:** This research has been funded by Scientific Research Deanship at University of Hail Saudi Arabia through project number RCP-24 028.

**Competing interests:** The authors have declared that no competing interests exist.

the interactions between proteins and ligands. The study concluded that **S904-0022** exhibited substantial therapeutic potential and requires additional experimental exploration as a potential NDM-1 inhibitor.

## 1. Introduction

β-lactamase enzymes have becoming subject of important due to their role in the emergence of antibiotic-resistant bacteria, which presents a substantial risk to the public health worldwide. [1,2]. Out of these enzymes, New Delhi Metallo-β-lactamase-1 (NDM-1) has attracted significant interest due to its ability to break down a broad spectrum of β-lactam antibiotics thereby rendering them ineffective [3]. NDM-1-producing bacteria have been detected in several regions of the globe, resulting in severe illnesses that are challenging to manage using traditional medicines [4,5]. The pressing requirement for novel therapeutic approaches has stimulated interest in investigating natural substances as possible inhibitors of NDM-1.

Pharmacologically active compounds have been abundantly found in natural products for an extended period, which has served as the foundation for numerous contemporary medications [6,7]. Their biological activities and structural diversity render them promising candidates for drug discovery, which includes the pursuit of novel antimicrobial agents. In this context, the identification and optimization of potential natural product inhibitors against NDM-1 can be significantly improved through the integration of computational approaches such as molecular docking, quantitative structure-activity relationship (QSAR) modeling, and molecular dynamics (MD) simulations.

Based on the chemical structure of compounds, QSAR modelling facilitates the screening of extensive libraries of natural products to identify those with potential inhibitory activity against NDM-1. This enables the prediction of biological activity. The exploration of the binding interactions between these compounds and the NDM-1 enzyme is facilitated by molecular docking simulations, which provide insights into their binding affinities and modes of action. By evaluating the stability and dynamics of the protein-ligand complexes over time, molecular dynamics simulations further refine these findings, thereby guaranteeing the identification of inhibitors that are both effective and robust.

Previous studies on NDM-1 (New Delhi Metallo-β-lactamase-1) has shown that this enzyme and its variations has a high level of efficacy in breaking down nearly all β-lactam antibiotics, resulting in substantial resistance to many drugs in bacteria [8–10]. There is currently no known way to effectively stop the activity of NDM-1 or other types of metallo-β-lactamases. This emphasizes the immediate requirement for new and innovative treatment approaches. Expanding upon this basis, a recent work employed a multi-step virtual screening method for the identification of non-β-lactam inhibitors against NDM-1 [11,12]. Previous studies have employed computational and experimental approaches to identify potential NDM-1 inhibitors, utilizing methods such as molecular dynamics simulations, isothermal titration calorimetry (ITC), and microbiological assays to assess binding affinity, stability, and enzymatic inhibition

[13,14]. These studies have demonstrated that certain inhibitors effectively reduce β-lactamase activity and enhance antibiotic susceptibility. Building upon this foundation, our study employs an advanced in-silico screening approach to identify novel natural product-derived inhibitors with therapeutic potential.

The objective of this study is to identify and investigate natural product compounds with potential inhibitory activity against NDM-1, aiming to discover novel leads for antimicrobial development. To address the critical challenge of antibiotic resistance posed by NDM-1-producing bacteria, we employed an integrated computational approach involving molecular docking, QSAR modeling, and molecular dynamics (MD) simulations. Further, principal component analysis (PCA), free energy landscape (FEL) and molecular mechanics generalized born surface area (MM/GBSA) were also used to understand the stability and favorable interaction of the identified compound. Thus, this study identified an inhibitor candidate for New Delhi Metallo-β-lactamase-1 (NDM-1).

## 2. Methodology

### 2.1. Protein structure prediction and molecular docking

The crystallized structure of the New-Delhi metallo-β-lactamase-1 (NDM-1) protein attached to meropenem (0RV) was obtained from the protein data bank (PDB) ID: 4EYL [4,15]. The already known inhibitor (0RV) was used as a control for further studies. The natural based product library was downloaded, which includes 4,561 natural product compounds from ChemDiv(https://www.chemdiv.com/catalog/focused-and-targeted-libraries/natural-product-based-library/ date: 15 June 2024).

### 2.2. ML-based QSAR model

The machine learning(ML)-based QSAR study used six regression models: linear regression, random forest regression, bayesian ridge regression, decision tree regression, support vector regression, and gradient boosting regression. Then, the ChEMBL database was then searched [14] (https://www.ebi.ac.uk/chembl/) for compounds needed for the prediction models. The keyword "New Delhi metallo-β-lactamase-1" was used to search in the ChEMBL database, which led to the identification of 43,867 target IDs linked to their binding compounds and activity levels listed in S1 Table. Compounds with a "MIC" activity score were specifically targeted and selected, which were 26,489 IDs used for further model building. These compounds were chosen based on their activity unit, namely those with a measurement of µg/mL. Prior to selection, any duplicate compounds and empty entries were removed. Subsequently, the RDKit software [16] was used to confirm the 3D descriptors (MACCS keys) of these substances. The MACCS (Molecular ACCess System) keys were selected for their well-established function in cheminformatics, which provides a standardized and interpretable set of 166 structural features. These keys are an efficient and dependable method for capturing critical molecular characteristics that are pertinent to biological activity. Consequently, they are appropriate for virtual screening tasks and compound comparison in drug discovery. Initially, the MIC values of the compounds were standardized by transforming them into logarithmic values using base 10. The length of the SMILES sequence is also ascertained. The target variable is determined by dividing the log10(MIC) value by the length of the SMILES sequence. The normalized target value can be calculated using the formula

$$final\ predicted\ MIC\ =\ antilog10(seq\_len * pred\_MIC) \tag{1}$$

The training-test dataset has been listed in the S2 Table. 30% of these compounds were allocated as a test set, while 70% were allocated for training the models. The estimated value used to validate the trained model is R2, also known as the coefficient. The trained model was used to perform a quantitative structure-activity relationship (QSAR) analysis on 4,561 natural product compounds and a control ligand to determine their activity levels.

$$final\ predicted\ EC50\ =\ antilog10(seq\_len * pred\_EC50) \tag{2}$$

As a result of the QSAR analysis, the compounds that demonstrated better activity levels than the control were selected for docking with control.

## 2.3. Virtual screening

A grid box surrounding the residues of the native ligand was generated using AutoDockTools [17], followed by the construction of a configuration file. AutoDockTools defined the grid box around the co-crystallized ligand residues with a 6 Å margin. The grid center coordinates were set at (2.19, −40.58, 2.22) for the x, y, and z axes, respectively, with dimensions of 20 Å (x-axis), 16 Å (y-axis), and 16 Å (z-axis), optimizing space to accommodate ligand flexibility while maintaining computational efficiency. The 3D structures of all compounds were obtained from ChemDiv and minimized using the OpenBabel tool [18] with the MMFF94 force field in 2500 steps [19] to ensure conformational stability. Specific docking parameters were determined to optimize the results. Docking simulations were performed using AutoDock Vina [19] with an exhaustiveness level of 10, balancing accuracy. 10 binding poses were generated per ligand to capture diverse interaction modes. The exhaustiveness and number of poses were chosen based on prior studies and empirical testing to provide reliable binding predictions without excessive runtime. The following parameters were used to conduct docking in AutoDock Vina: ligand flexibility was enabled, enabling the compounds to adopt a variety of conformations during docking; the scoring function used by Vina estimated binding affinities based on an empirical free energy model; and 10 docking poses were generated per compound to capture a spectrum of binding orientations. [20]. The normalized binding scores were ranked based on these values to enable comparative ranking across ligands, following the equations below.

$$average \ = \ Average \ of \ (Binding \ Energy \ / \ Top \ Binding \ Energy) \tag{3}$$

$$Normalized \ Score \ = \ Top \ Binding \ Energy \ * \ average \tag{4}$$

## 2.4. Tanimoto similarity and clustering

Compounds that exhibited superior interaction compared to control were chosen for further screening based on their best binding energy. Additional comparisons were conducted on the selected compounds using the RDKit tool [16] and Tanimoto similarity. The k-means function from the cluster module in the sklearn library of Python [21] was used for clustering. The graphs were made using the matplotlib tool in Python [22]. The number of clusters was determined to be k = 3 in order to maintain a balance between interpretability and diversity. This decision was made after preliminary analysis revealed that three distinct chemical subgroups had emerged based on similarity scores. This value avoided overfitting or fragmenting the dataset, thereby achieving meaningful separation. The centroid of each cluster, which represented the whole cluster, was recovered after the clustering process. The compounds of the centroids were employed for molecular-dynamics-simulation.

## 2.5. Molecular-dynamics-simulation (MDS)

The study employed the Gromacs 2022.4 software [23] to perform a comprehensive 300 ns molecular dynamics (MD) simulation to analyse interactions inside a protein-ligand complex. The CHARMM36 force field (Chemistry at HARvard Macromolecular Mechanics force field version 36) [24] was used to determine the molecular interactions between the target protein and the ligands. The ligands, including our target molecule and a control inhibitor, underwent force-field parameter creation and optimization using the CGneFF (CHARMM General Force Field) server [25]. The implementation of the Particle Mesh Ewald (PME) approach [26] was done to model electrostatic interactions over a certain distance accurately. Using the TIP3P water model, the complex was placed in a dodecahedron water box and neutralized with Na⁺

and Cl⁻ ions [27]. Before molecular dynamics simulation, 50,000 steepest descent steps were performed to resolve steric clashes and prepare the system. For system stability we applied LINCS (Linear Constraint Solver) algorithm for limits on bond lengths [28]. After energy minimization, the system was gradually heated from 0 K to 310 K using a 2 femtosecond (fs) timestep over a 100 picosecond (ps) simulation under the NVT ensemble (constant number of particles, volume, and temperature). This heating step served as the annealing phase. Subsequently, the system was equilibrated at 310 K and 1 atmosphere in the NPT ensemble (constant number of particles, pressure, and temperature) for 1 nanosecond (ns), using the same 2 fs timestep to stabilize temperature and pressure prior to the production molecular dynamics simulation. Throughout the 300 ns production nphase of the simulation, the coordinates of the structure were constantly recorded at intervals of 10 picoseconds. Velocity scaling [29] was applied to maintain a constant system temperature during the simulation, while pressure control was achieved using the Parrinello–Rahman coupling method [30]. The post MD analysis was performed on the visual platform called "Analogue" developed by Growdea Technologies [31,32] (https://growdeat-ech.com/Analogue/). The conformational-stability and variations were assessed by employing metrics such as root mean square deviation (RMSD) and root mean square fluctuation (RMSF). The hydrogen bonding patterns within the protein–ligand complexes were analyzed using the gmx hbond module in GROMACS, which calculates the number, duration, and occupancy of hydrogen bonds over the simulation trajectory to provide insights into the stability and dynamic interactions of the complexes.

**2.5.1. PCA (Principal Component Analysis) and FEL (Free Energy Landscape).** The trajectory was preprocessed for principal component analysis by eliminating the periodic boundary condition. The covariance matrix was calculated using the Gmx_covar component of GROMACS. The covariance matrix is used to define the correlation between the atomic fluctuations of the protein-ligand pair. The gmx analysis function was used to calculate the eigenvalues and eigenvectors of the covariance matrix. The GROMACS tool 'gmx anaproj' was used to calculate the PC (Principal Component) coordinates for all frames.

The investigation of the transitional state, represented by the obstacles on the free energy landscape (FEL), and the equilibrium state, represented by the lowest points on the free energy landscape, can yield a significant information regarding biological-system phenomena such as the recognition, aggregation, and folding of biomolecules [33]. In order to calculate the free energy landscape, the energy distribution was estimated utilizing Equation (5).

$$\Delta G(X) = -k_BT\ln P(X) \tag{5}$$

The Boltzmann constant is represented as $k_B$, the Gibbs free energy is symbolized by G, the reaction coordinates are indicated as X, and the system's probability distribution along the reaction coordinates is denoted by P(X).

**2.5.2. MM/GBSA.** Methods of Generalised Born Surface Area and Molecular Mechanics are combined in the MM/GBSA process. By way of predicting the free energy of binding, the process utilised to ascertain the intensity of bonds of the ligands and the receptor, which is frequently a protein, is denoted. Using the MM-GBSA method and the GROMACS module (gmx_MMGBSA) [34,35], the free energy of binding in the complex was computed during the last 20 nanoseconds of the simulation. Utilizing the MM/GBSA method, the analysis of binding free energy was performed. Assuming the subsequent equation, the MM/GBSA calculation was executed:

$$\Delta G = G_{complex} - [\, G_{receptor} + G_{ligand} \,] \tag{6}$$

$$\Delta G_{binding} = \Delta H - T\Delta S \tag{7}$$

$$\Delta H = \Delta G_{GAS} + \Delta G_{SOLV} \tag{8}$$

$$\Delta G_{GAS} = \Delta E_{EL} + \Delta E_{VDWAALS} \tag{9}$$

$$\Delta G_{SOLV} = \Delta E_{GB} + \Delta E_{SURF} \tag{10}$$

$$\Delta E_{SURF} = \gamma.SASA \tag{11}$$

In Equation 6, the total free energies of the protein-ligand complex ($\Delta G_{complex}$), the unbound protein ($\Delta G_{receptor}$), and the ligand ($\Delta G_{ligand}$) in the solvent are denoted as $\Delta G_{ligand}$, $\Delta G_{receptor}$, and $\Delta G_{ligand}$, respectively. Equations in range of (7–11) illustrated the association between different components (solvation-free-energy change ($\Delta G_{SOLV}$), conformational-entropy change (-T$\Delta$S), enthalpy change ($\Delta$H), and gas-phase-energy differential ($\Delta G_{GAS}$)). The accessible surface area to the solvent was designated as the solvent-accessible surface area (SASA). On the other hand, the symbol $\gamma$ is used to represent the solvent surface tension. The $\Delta E_{VDWAALS}$ was used to represent the van der Waals energy alterations, while $\Delta E_{EL}$ represent electrostatic energy variations. $\Delta E_{GB}$ used for solvation energies changes in polar molecules, while $\Delta E_{SURF}$ used for nonpolar substances solvation energies.

## 3. Results

### 3.1. Protein structure

New Delhi Metallo-β-lactamase-1 (NDM-1) was selected as the target of this study due to its important involvement in antibiotic resistance. NDM-1 is an enzyme that gives resistance to a wide range of β-lactam antibiotics, including carbapenems, which are frequently regarded as the final line of defense against multidrug-resistant bacterial infections. The PDB structure of NDM-1 (PDB ID: 4EYL) was selected that was bound to the known ligand hydrolyzed meropenem (0RV). Fig 1 depicts the specific amino acid residues that are involved in binding to the hydrolyzed form of meropenem in the crystal structure of NDM-1. These residues were visualized using the PyMOL program [36]. The residues Val73, Ala74, Ser75, Trp93, Asp124,

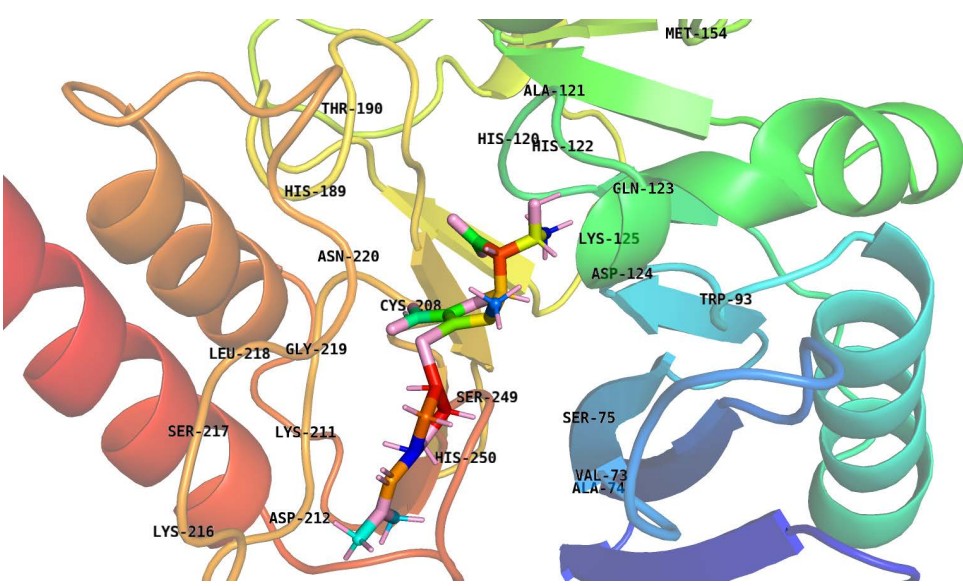

**Fig 1. Binding residues surrounding the hydrolyzed meropenem bound to crystal structure of NDM-1.** The binding residues were identified using the PyMOL tool and are defined as those situated within a 6 Å radius of the known ligand.

Lys125, Gln123, His122, His120, Ala121, Thr190, His189, Asn220, Gly219, Leu218, Ser217, Lys211, Lys216, Asp212, Ser249, His250 and Cys208, which are located within a 6 Å radius of the ligand, play a critical role in the binding interactions. This comprehensive visualization facilitates comprehension of the interaction between NDM-1 and β-lactam antibiotics such as meropenem, which is crucial for the development of potent inhibitors to counter antibiotic resistance.

Previous studies showed that the Zn1 ion made covalent bonds with the His120, His122, and His189 residues, whereas the Zn2 ion created covalent bonds with the Asp124, Cys208, and His250 residues. Val173, Ile35, Leu65, Met67, and Trp93 formed a large hydrophobic binding surface, facilitating interactions with β-lactam R groups [37–39]. Lys211 and Asn220 help recognize and hydrolyze substrates [39]. Hydrogen interactions were found between oxygen atoms next to hydrophobic β-lactam R groups and Gln123 and Asp124 [37]. Trp93, Gln123, Asp124, and His250 were found to be involved in the development of hydrophilic pores that bind to substrates [40]. These residues were located within the binding pocket where the known ligand, 0RV. These residues were later used during the virtual screening using molecular docking. The figure offers crucial insights for drug design and the creation of new inhibitors that can enhance the effectiveness of β-lactam antibiotics by showing the spatial organization and interactions of these residues.

### 3.2. ML-based QSAR model

In this study, screening was performed with quantitative structure-activity relationship (QSAR) models based on machine learning. These computational tools are extremely useful in predicting the biological activities or properties of chemical substances. By establishing mathematical relationships between molecules' structural properties or descriptors and their observable activities or qualities, QSAR models use machine learning to find possible inhibitors and aid the drug design process [41].

Machine learning-based QSAR models are widely employed in drug discovery and chemical safety evaluation. Machine learning QSAR algorithms can predict the biological activity of enormous chemical libraries, identifying compounds that may be active against a certain target [42]. In this work, the "Targets'' section of the ChEMBL database was searched using the keyword "New Delhi metallo-β-lactamase-1'', leading to the identification of an individual target ID (CHEMBL350) with 43,867 molecules with activity. The "MIC" was chosen because it contained 26,489 chemicals from the "CHEMBL350" target. After deleting duplicates and blanks, only 12,107 compounds with an activity unit in μg/mL were considered for QSAR training and testing purposes. The 3D descriptors (MACCS keys) of these compounds were then calculated using the RDKit software. The selected MACCS keys were chosen due to their ability to provide a fixed-length, interpretable fingerprint that is composed of 166 predefined structural features that are frequently encountered in bioactive molecules. This renders them especially advantageous for duties such as clustering, similarity analysis, and rapid screening. MACCS keys are computationally efficient and are widely used in cheminformatics due to their balance between chemical relevance and simplicity, whereas other fingerprinting methods, such as Morgan (circular) fingerprints, ECFP (Extended Connectivity Fingerprints), or RDKit topological fingerprints, capture more detailed structural or connectivity-based information. Additionally, their standardized character enables them to be compared across datasets and studies, rendering them a practical option for this analysis. Specifically, 12,107 compounds with minimum inhibitory concentration (MIC) values (in μg/mL) were retrieved from the ChEMBL database (Target ID: CHEMBL350) after removal of duplicates and missing values. MIC values were converted to $\log_{10}$ scale to normalize data and ensure consistency. The log scale standardizes the scales of all compounds, maintaining consistency throughout model training and eliminating bias. Similarly, to predict the MIC values using machine learning-based QSAR models, a dataset with information on chemical compounds and their experimental MIC values is required. 3D descriptors (MACCS keys) were then constructed to represent the chemical structures. These descriptors and MIC values were used to create a machine learning model, which was then validated using cross-validation methods. 70% of the compounds were set aside for model training, with the remaining 30% for testing. Furthermore, the coefficient of determination (R2) value was calculated for model validation; the $R^2$ reflects the degree of correlation between predicted and observed data and was used to assess the model's

effectiveness. The examination of all models yielded promising results for random forest (RF) model, with the lowest $R^2$ value reached being 0.70 and the lowest mean squared error (MSE = 0.0002), as shown in Table 1. This shows that, among the models considered, the random forest (RF) model had the best predictive performance, as evidenced by its best $R^2$ value.

Later, the trained model was used to perform QSAR on 4,561 natural product molecules and a control ligand (0RV) to evaluate their relative activity levels. The MIC of the compounds was estimated using the random forest (RF) model and ranked based on the final anticipated MIC values. The QSAR analysis revealed 699 compounds showed higher activity levels than the control (0RV) as listed in the S3 Table. Thus, these 699 screened compounds were further selected for another round of screening using molecular docking along with the control (0RV).

### 3.3. Molecular docking

In the current investigation, the protein 3D crystal structure of NDM-1 (PDB: 4EYL) was used in the AutodockTools with grid box information and docked to the 699 screened compounds from the ML-based QSAR model. The binding site residues surrounding the known ligand (0RV) were used to create the grid box. The known ligand was also docked and used as a cutoff point for calculating the compounds' binding scores. The co-crystallized ligand (ORV) was re-docked into the active site of the NDM-1 protein (PDB ID: 4EYL) using the same docking parameters that were used in the screening study to validate the docking protocol. The superimposed S1 Fig verified that the docked pose was closely aligned with the original crystal structure of the ligand. The docking setup's reliability and accuracy are clearly demonstrated by the minimal deviation between the native (green) and re-docked (yellow) conformations. The virtual screening results are substantiated by this validation phase.

The binding scores of all 699 compounds, as well as the control, were then normalized and ranked accordingly as listed in the S5 Table. The normalized score is a relative measure of the binding affinity of each compound in comparison to the top-performing ligand, which aids in the reduction of bias resulting from absolute energy differences. This ranking method guarantees a consistent and equitable comparison of all 699 compounds, particularly when resource constraints restrict the number of docking experiments that can be conducted. It enables the efficient prioritizing of candidates who are likely to have robust interactions. Out of 699 substances examined, 299 had a higher normalized binding score than the control, as shown in S4 Table. The control had a binding scores −6.8 kcal/mol and these 299 compounds had binding scores in the range < −6.8 to −9 kcal/mol. According to the normalized scores, 299 compounds exceeded the performance of the control ligand (ORV, –6.8 kcal/mol; normalized score: –6.31). The top molecule, D751-0254, had a binding energy of –8.4 kcal/mol and a normalized score of –7.89, whereas the final selected compound, S943-0430, demonstrated –7.0 kcal/mol and a normalized score of –6.32. These 299 compounds were prioritized for subsequent study, encompassing clustering and molecular dynamics simulations.

Several previous studies have reported NDM-1 inhibitors with varying binding affinities. One study identified 18 compounds that closely bound to the active site of NDM-1, all with binding scores below −8.8 kcal/mol [12]. Another study

**Table 1. Validation metrics for the trained model coefficient of determination ($R^2$) and mean squared error (MSE) values.**

| Model | R2 | MSE |
|---|---|---|
| Bayesian Ridge | 0.16 | 0.0005 |
| Linear Regression | 0.16 | 0.0005 |
| **Random Forest regressor** | **0.70** | **0.0002** |
| Decision tree regressor | 0.37 | 0.0002 |
| Support Vector Regression | −0.16 | 0.0007 |
| Gradient Boosting Regression | 0.53 | 0.0003 |

shortlisted five compounds as potential inhibitors, with binding scores ranging from −8.32 to −8.79 kcal/mol [9]. Similarly, a separate investigation reported binding energies of −7.7 kcal/mol and −8.6 kcal/mol for its identified NDM-1 inhibitors [43]. Another docking study found that the most promising compounds had docking scores below −8.29 kcal/mol [44]. Furthermore, a study on NDM-1 inhibitors identified the top five compounds with binding scores between −8.42 kcal/mol and −8.83 kcal/mol [13]. Additionally, among various antibiotics and inhibitors examined, meropenem exhibited the most favorable outcome with a docking score of −9.24 kcal/mol, whereas captopril had a weaker binding score of −6.51 kcal/mol [45]. Another computational study selected six ligands with AutoDock Vina scores ≤ −8.0 kcal/mol for further analysis, including MCULE-1996250788-0-2, MCULE-8777613195-0-12, MCULE-2896881895-0-14, MCULE-5843881524-0-3, MCULE-4937132985-0-1, and MCULE-7157846117-0-1 [46].

In comparison, our study identified 299 compounds with binding scores ranging from −6.8 kcal/mol to −9.0 kcal/mol, demonstrating similar or superior binding affinities relative to previously reported inhibitors. The control compound in our study had a binding score of −6.8 kcal/mol, suggesting that a substantial number of screened compounds exhibited stronger binding interactions with NDM-1. These results highlight the potential of our identified compounds. Further, these compounds were then selected for tanimoto similarity and clustering analysis.

### 3.4. Tanimoto similarity and clustering

The 299 compounds were analysed for tanimoto similarity estimates. Fig 2a depicts a heatmap illustrating the degree of similarity between the chemicals. The diagonal yellow line from the top left corner to the bottom right indicates that each molecule was compared to itself, yielding the highest similarity score of 1.0, as shown by the colour scale on the right. The colour scale runs from purple (0.0) to yellow (1.0), showing considerable similarities. The heatmap is mostly purple, indicating that most molecules share little resemblance. Some green to yellow squares are scattered,

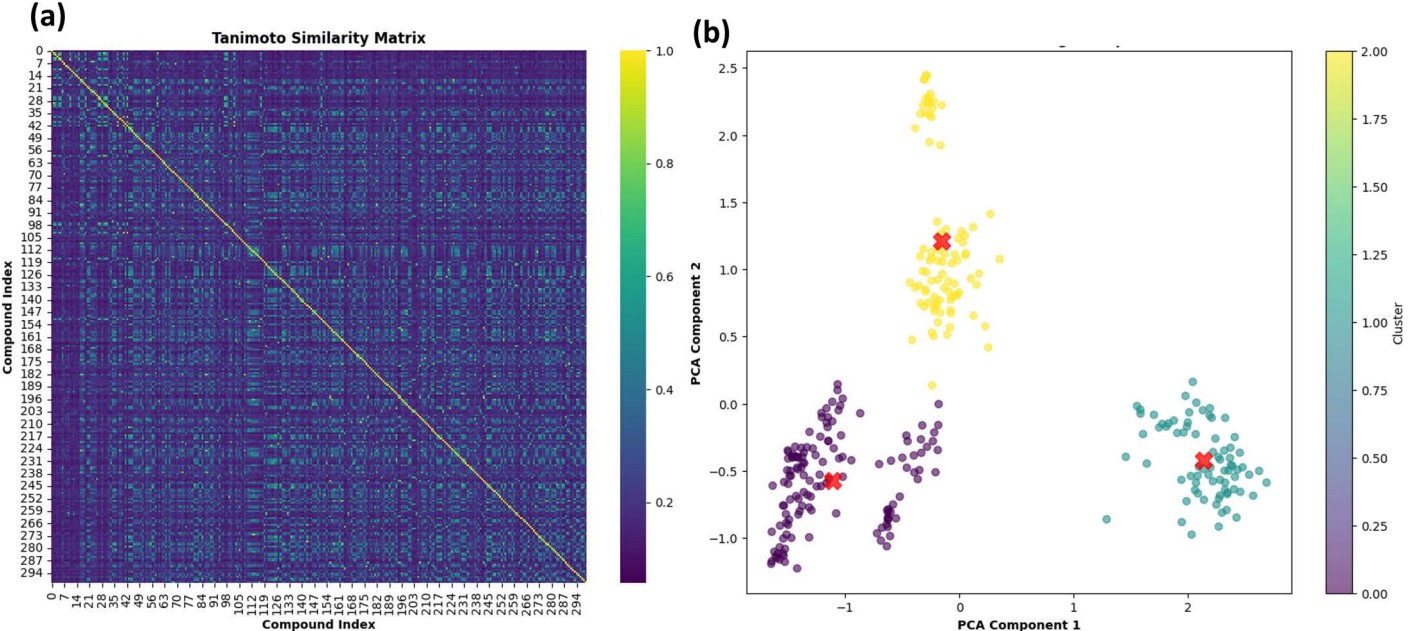

**Fig 2. (a) The Tanimoto similarity matrix shows compound similarity ratings. Each cell shows the similarity between two compounds, with yellow denoting high and purple low. Each compound is self-similar on the diagonal. (b) PCA K-means clustering of peptides. The first two main components from PCA are used to cluster peptides. K-means clusters are colored. The cluster centroids, shown by red stars, are central in PCA space.**

indicating locations where specific molecules have more similarities. The clustering algorithm then grouped the chemicals based on their similarity in a multidimensional space. Furthermore, Fig 2b depicts the k-means clustering plot, with centroids indicated by a red X. Clustering revealed three separate clusters, two of which constituted a closed cluster, indicating that a compact cluster had less variability among data points within that group. The clusters are listed in the S6, S7 and S8 Tables, respectively. The homogeneity of comparable molecules can be quite useful in applications that require predicting their behavior or qualities, because close grouping suggests reliable and consistent properties. Thus, the molecule at the centroids is the genuine representative of each cluster. The three centroids (**S721-1034, S904-0022,** and **N118-0137**), each representing a molecule from a cluster, were chosen for the molecular dynamics simulation. The selection of each molecule was determined by its proximity to the centroid within its cluster, thereby guaranteeing that it accurately reflects the structural and chemical diversity of the cluster to which it belongs. The exact structural representation of these molecules is provided by the SMILES strings as shown in Table 2, which facilitates additional number of members in the cluster formed. Here, Fig 3 represents the 3D structure of the three compounds and the control.

**Table 2. SMILES of the compounds (three centroids) found from the k-mean clustering.**

| Compound | SMILES | Distance from Centroid | Cluster | Total Member |
|---|---|---|---|---|
| S721-1034 | C1(NN(CC(=O)N2[C@H](C(=O)NC3CC3)C[C@@H](C2)OC2CNCCC2)C(C1)C)C(F)(F)F | 0.034842611 | 1 | 134 |
| S904-0022 | C1(N2C(NN1)CN(C(=O)C1C(NC(CC1)OC)OC)CC2)[C@H]1N(C[C@H](C1)O)C(C)C | 0.06141058 | 2 | 76 |
| N118-0137 | S(=O)(=O)(C1CN(NC1)CC)NC1CCN(C(=O)N2C[C@@H]3[C@@H]4N(C(=O)CCC4)C[C@@H](C2)C3)CC1 | 0.042962854 | 3 | 90 |

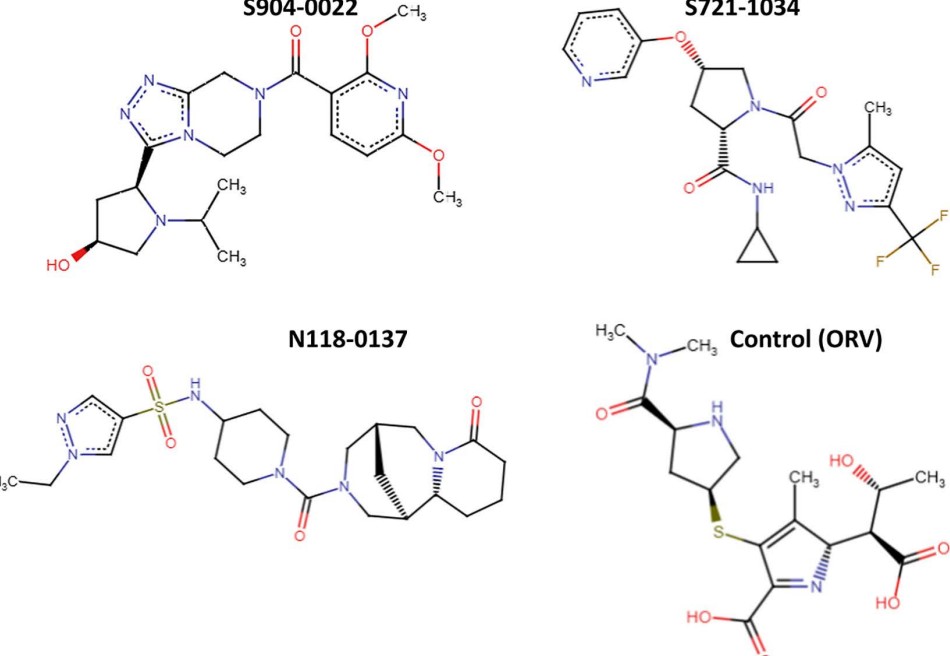

**Fig 3. 2D chemical structures of N118-0137, S721-1034, S904-0022 and control (ORV).**

### 3.4. Molecular dynamics simulation

**3.4.1. RMSD.** The Fig 4a depicts the root mean square deviation (RMSD) of the protein Cα atoms over a period of time in molecular dynamics (MD) simulations. It shows the RMSD values for a control group and three distinct ligands (**N118-0137, S721-1034, S904-0022**). Significant fluctuations in protein RMSD can suggest ligand-induced conformational changes or local destabilization, which may influence protein function or binding affinity. Such changes are important to interpret in the context of ligand stability and binding mechanisms during molecular dynamics simulation [47,48].

The RMSD of the protein, when it is attached to the control, exhibits consistent changes within the range of 0.2 to 0.25 nm throughout the whole simulation period. This suggests that the protein structure remains stable. The RMSD of the protein when bound to **N118-0137** exhibits a comparable pattern to the control, as indicated by the RMSD values falling within the range of around 0.2 to 0.3 nm. This suggests that **N118-0137** does not have a substantial destabilising effect on the protein structure. The RMSD values for **S721-1034** are similar to the control and **N118-0137**, oscillating between 0.25 to 0.35 nm. This suggests a mild effect on the stability of the protein. The protein bound to **S904-0022** demonstrates the marginally high RMSD values, which range from 0.3 to 0.4 nm. These values suggest that there is a greater amount of structural variability in this variant, potentially leading to a more pronounced effect on protein stability. The control and **N118-0137** demonstrate reduced RMSD values, indicating that the protein retains a stable structure under these conditions. On the other hand, **S904-0022** has elevated RMSD values, suggesting more pronounced alterations in structure and possible instability. The differing RMSD values for various ligands indicate distinct impacts on the structural stability of the protein. Ligands that result in higher RMSD values may trigger conformational alterations of the protein, whilst those with lower RMSD values have a comparatively smaller effect. The RMSD plot is a useful tool for evaluating the stability and structural changes of proteins in molecular dynamics simulations.

The RMSD of a control and three distinct ligands (**N118-0137**, **S721-1034**, **S904-0022**) is illustrated in Fig 4b during a 300 ns molecular dynamics simulation. Consistently remaining below 5 nm throughout the simulation, the control exhibits negligible RMSD fluctuations, indicating a stable reference structure with no substantive conformational alterations. The **N118-0137** ligand demonstrates substantial RMSD values, which initially peak at approximately 10 nm and then gradually increase to approximately 25 nm. This suggests that the ligand has a propensity to exit the protein's binding site. This observation is further corroborated by S2 Fig, which illustrates the structural modifications in **N118-0137** at various time intervals (0 ns, 100 ns, 200 ns, and 300 ns). The ligand is firmly anchored within the active site at 0 ns (S2a Fig). However, as the simulation advances (S2b-d Fig), the ligand gradually displaces from the protein's surface, thereby verifying weak and unstable interactions.

Similarly, the RMSD of **S721-1034** showed increase, with values as high as 40 nm, which further exacerbates its instability in the binding site. S3 Fig illustrates the structural transitions of **S721-1034** during the molecular dynamics' simulation. S3a Fig illustrates the ligand's initial binding within the active site. Nevertheless, **S721-1034** begins to lose contact with the protein by 100 ns (S3b Fig), and the ligand is substantially displaced by 200 ns (S3c Fig). By 300 ns (S3d Fig), **S721-1034** is completely dissociated from the binding site, which supports the significant RMSD fluctuations and implies that the binding is not stable.

Further, the RMSD of the control was compared with the ligand **S904-0022** as shown in Fig 3c. The RMSD of the control exhibits a constant and steady pattern during the simulation, with values consistently between 0.5 nm to 1 nm. This suggests that the control structure remains stable without undergoing major conformational changes. The ligand **S904-0022** exhibits slightly higher RMSD values in comparison to the control, with RMSD of 2 nm. The RMSD displays prominent peaks reaching 4 nm, especially at the beginning of the simulation, suggesting notable alterations in molecular structure and flexibility. These alterations may have implications for the ligand's binding effectiveness and stability when interacting with the target protein. However, post 150 ns **S904-0022** shows constant RMSD suggesting stable binding with the protein.

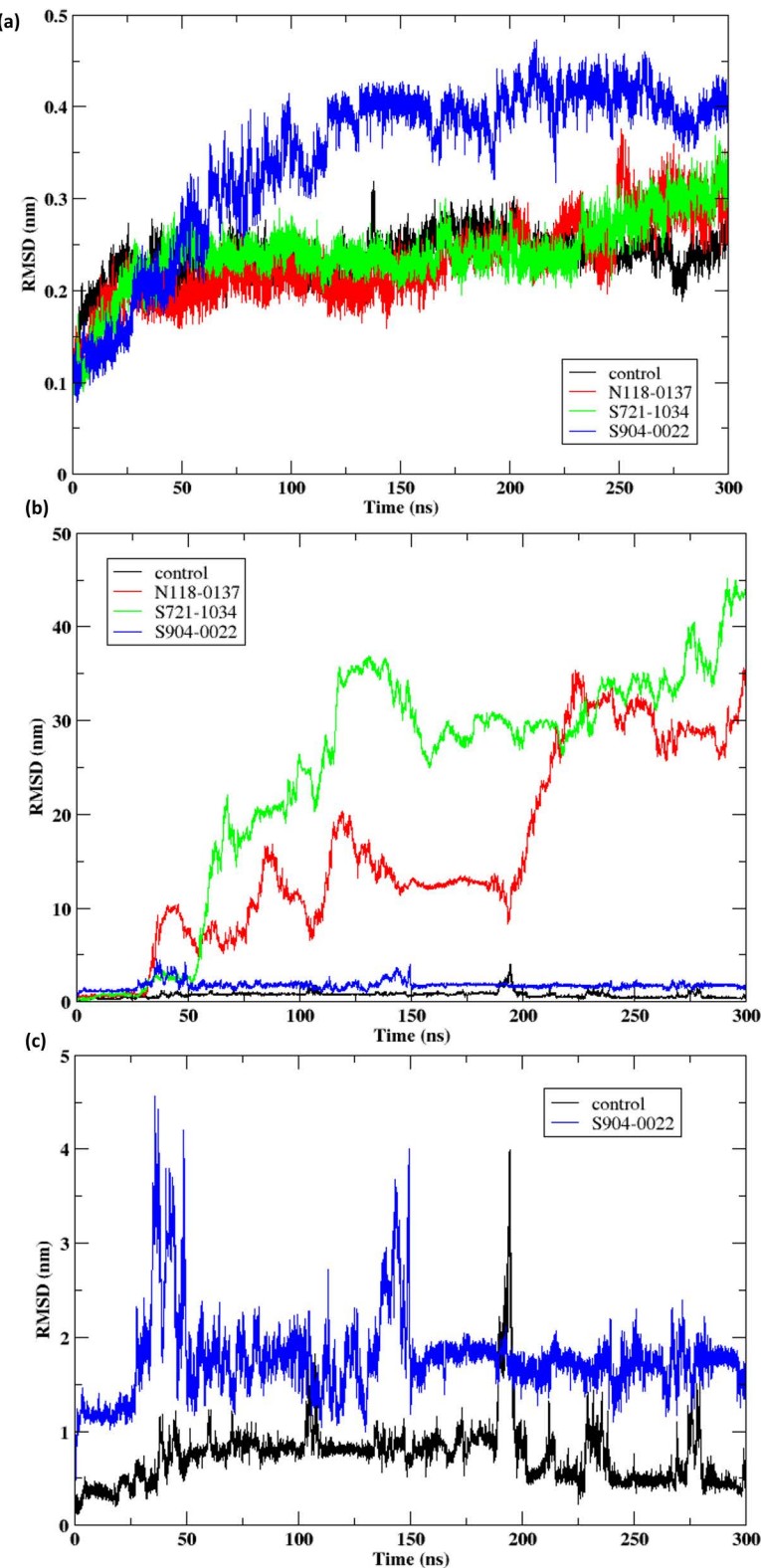

**Fig 4. Post molecular dynamics simulation analysis of the NDM-1 when bound to the compounds (a) RMSD of the protein Cα atoms when bound to the compounds N118-0137, S721-1034, S904-0022 and control (ORV) (b) RMSD of the ligands during the 300 ns molecular dynamics simulation (c) RMSD of the control (ORV) and S904-0022.**

Numerous prior studies have examined the stability of NDM-1 inhibitor complexes through molecular dynamics (MD) simulations. A study indicated that the NDM-1-ZINC05683641 complex attained equilibrium at 100 ns, with RMSD values stabilizing at approximately 0.2 nm, signifying structural stability. The last 60 nanoseconds of the simulation were utilized for additional investigation. [12]. Another study examining a different NDM-1 inhibitor observed high RMSD deviations (0.08 nm to 0.12 nm) during the first 5 ns, followed by stabilization. The RMSD fluctuations between 0.10–0.18 nm indicated steady-state dynamics of the protein backbone. Additionally, the ZINC84525623 ligand initially stabilized after 3 ns, but exhibited fluctuations between 0.4–0.6 nm, likely due to its large size and repositioning within the NDM-1 active site cavity [9]. It is essential to note that RMSD values, while informative, should not be interpreted in isolation. Moderate RMSD values can still indicate stability, especially for ligands exhibiting dynamic repositioning within the active site, as seen with larger molecules like ZINC84525623. Another molecular dynamics study analyzed multiple NDM-1-ligand complexes, reporting stable protein backbone RMSD trajectories for M75 (0.093 nm), M1 (0.102 nm), and M17 (0.103 nm), while M21 (0.129 nm) and M61 (0.134 nm) displayed less stable trajectories [13]. In a separate study, the metallo-β-lactamase inhibitor captopril underwent three independent 100 ns molecular dynamics simulation runs, and RMSD values were used to assess binding stability. Most inhibitors initially peaked at 0.10 nm, except meropenem, which remained more stable at 0.075 nm [45].

Compared to these studies, our findings suggest that **S904-0022** demonstrates a stable interaction with NDM-1, despite slightly higher RMSD values than the control. While it exhibits minor fluctuations (0.30–0.40 nm), these values remain lower than the fluctuations observed for certain previously reported inhibitors, indicating a structurally stable complex. Importantly, S904-0022 maintained its interaction within the active site without significant conformational drift or dissociation, highlighting the potential of this compound to sustain meaningful interactions over a biologically relevant timescale. Overall, **S904-0022** remains within the stability range of successful inhibitors reported in the literature, supporting its potential as a promising NDM-1 inhibitor.

These support the hypothesis that S904-0022 not only fits well within the active site but also modulates the protein conformation in a functionally relevant manner. This is particularly promising given that inhibitors must often compete with flexible β-lactam antibiotics in the same pocket. The combination of persistent interactions, stable RMSD profile, and favorable binding free energy collectively suggest that S904-0022 mimics the characteristics of effective NDM-1 inhibitors reported in the literature, while also offering a distinct chemical scaffold for further optimization.

**3.4.2. RMSF.** The Fig 5 displays the root mean square fluctuation (RMSF) of protein residues during a 300 ns molecular dynamics (MD) simulation for a control and three ligands (**N118-0137, S721-1034, S904-0022**). The control has consistently low RMSF values for the majority of residues, suggesting minimal variation and a high level of stability. **N118-0137** displays marginally elevated RMSF values in comparison to the control, indicating mild variations at certain residues. **S721-1034** exhibits a comparable pattern to that of **N118-0137**, although with significantly more variations observed at certain residues. **S904-0022** data displays the most elevated RMSF values, specifically at the N-terminus and C-terminus. This suggests notable fluctuations and probable instability in these areas. The control exhibits reduced RMSF values, suggesting enhanced stability of the protein structure in the presence of these ligands. On the other hand, **S904-0022** has elevated RMSF values, indicating greater flexibility of residues and the possibility of conformational changes in the protein structure. Thus, this may indicate the ligand **S904-0022** may have triggered the flexibility of the protein while the other two compounds which moved out of the binding site of the protein does show any effect on the protein. RMSF analysis facilitates the comprehension of how various ligands impact the pliability and robustness of protein residues. This information is vital for the development of ligands that improve protein stability.

**3.4.3. Hydrogen bonds.** The Fig 6 presents the number of hydrogen bonds formed between the protein and ligand **S904-0022** compared to a control over 300 ns molecular dynamics simulation. The number of hydrogen bonds is plotted on the y-axis, while the simulation time in nanoseconds is plotted on the x-axis. The control shows varying numbers of hydrogen bonds throughout the simulation, with values ranging mostly between 1 and 5 hydrogen bonds. The presence

                                                                                                    

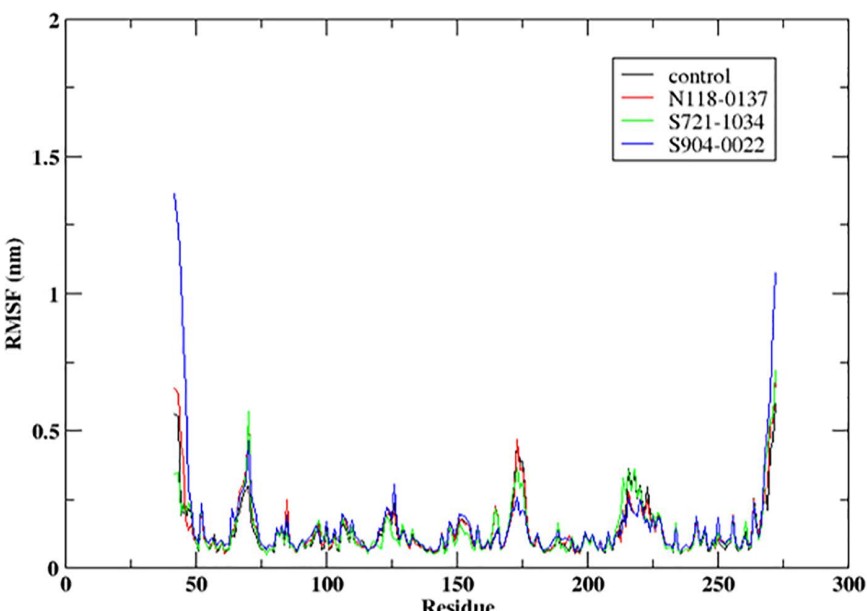

**Fig 5. RMSF of the protein Cα atoms during the 300 ns molecular dynamics simulation, plotted from MDS trajectories of NDM-1 bounded to ligands along with the used control ligand.**

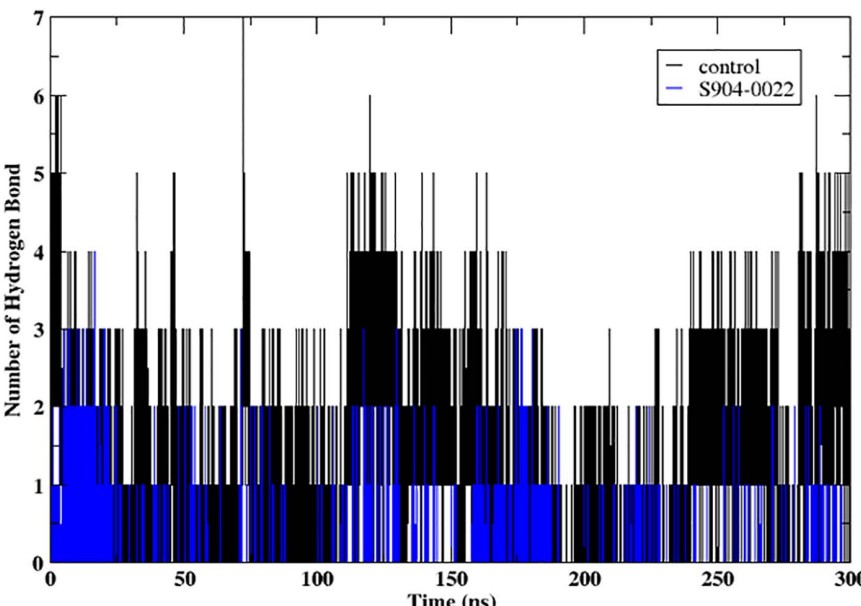

**Fig 6. Hydrogen bonds formed between protein and the ligands (control and S904-0022) for during the 300 ns molecular dynamics simulation.**

of **S904-0022** results in fewer hydrogen bonds compared to the control, generally maintaining around 1–3 hydrogen bonds throughout the simulation. Hydrogen bonds are critical for the stability and specificity of protein-ligand interactions. A higher number of hydrogen bonds typically correlates with stronger and more stable interactions. The reduced number of hydrogen bonds in the presence of **S904-0022** suggests that this ligand forms fewer hydrogen bonds with the protein

compared to the control. This could imply weaker or less stable interactions, potentially affecting the ligand's efficacy. Although S904-0022 forms primarily 1–2 hydrogen bonds throughout the 300 ns simulation, these interactions are maintained consistently, as shown in Fig 5, suggesting stable and persistent binding with the protein. While the number of hydrogen bonds is modest, prior studies have shown that sustained hydrogen bonding especially when accompanied by additional non-covalent interactions such as hydrophobic contacts and π–π stacking can significantly contribute to ligand stability and binding affinity [49].

**3.4.5. Interaction analysis.** The Figs 7 and 8 depict the 2D representation of interaction of the compounds when bound to the protein. In the initial state at 0 ns, the control compound established four conventional hydrogen bonds with His189, Asp124, Asn220, and His250. Additionally, it generated carbon hydrogen bonds with His120, Lys211, and Gly219. This indicates robust initial binding interactions, which enhance the stability of the ligand within the active region of the protein. At a pose of 100 ns, the control exhibited four typical hydrogen bonds, with the interacting residues being Lys211, His189, Leu218, and His250. These findings suggest a minor modification in the binding process, where Lys211 and Leu218 have significant importance. At the 200 ns pose, the control displayed typical hydrogen bonds with Asn220 and Ser217, as well as carbon hydrogen bonds with His250 and Gly219. This alteration implies that there are dynamic interactions occurring, in which various residues stabilize the ligand at different points in time. At the conclusion of the simulation (at 300 ns, in the final state), the control molecule established three conventional hydrogen bonds with Ser217, Leu218, and Asn220, as well as carbon hydrogen bonds with His250 and Lys216. The persistent existence of Asn220 and His250 in various positions underscores their crucial role in preserving the stability of binding.

In the initial state at 0 ns, **S904-0022** established a conventional hydrogen bond with His122, carbon hydrogen bonds with His189 and Gln123, and alkyl bonds with His250, Trp93, and Val73. This suggests a variety of early contacts, encompassing both hydrogen and hydrophobic bonds. At a pose of 100 nanoseconds, **S904-0022** exhibited a sole carbon-hydrogen bond with Glu152. The decrease in the quantity of bonds indicates a less stable interaction in comparison to the original state. At a posture of 200 ns, the ligand established both conventional and carbon hydrogen bonds with Gly222 and Glu152, respectively. This indicates a change in the individuals or entities with whom one interacts, which may suggest adaptations in the way the ligand binds. At a time of 300 nanoseconds in its final state, S904-0022 formed a typical hydrogen bond with Ser191 and an alkyl bond with Leu148. This final interaction pattern exhibits distinct binding kinetics in comparison to the control, characterized by a reduced number of hydrogen bonds and an increased number of hydrophobic contacts.

The control mostly depended on traditional hydrogen bonds for stability, while **S904-0022** predominantly utilized carbon hydrogen and alkyl bonds, which could result in a more diverse interaction pattern and potentially less stable binding. Residues such as His122 and Gly222 play a vital role at various time intervals for **S904-0022**, indicating a dynamic binding mode that adjusts over time. This study offers valuable insights into the kinetics of binding for both the control molecule and S904-0022, emphasizing the significance of particular residues and the type of interactions they engage in to determine the stability and effectiveness of binding.

A previous investigation revealed that ZINC05683641 exhibited significant interaction with critical residues (Ile35, Met67, Val73, Trp93, Cys208, Asn220, and His250) in the catalytic area of the active site, suggesting potential competing inhibitory activity. [12]. Another study found that ZINC84525623 impacts NDM-1's catalytic efficiency by building stable complexes through electrostatic and hydrophobic interactions [9]. A different study found that a compound (PHT427) interacts with key amino acid residues (Asn220, Asp124, and Gln123) [50]. The interaction with these residues (His250, Asn220, and Trp93) are essential across drugs for mediating interactions with NDM-1. These residues are frequently located near or within the catalytic core of NDM-1, which is critical for β-lactam hydrolysis. Disrupting interactions at these sites can potentially hinder the enzyme's ability to degrade antibiotics. Like the inhibitors addressed in prior studies, **S904-0022** in this investigation interact with important catalytic and structural areas of NDM-1, pointing to potential paths for inhibition. Notably, the overlap in binding residues between S904-0022 and known β-lactam antibiotics such as

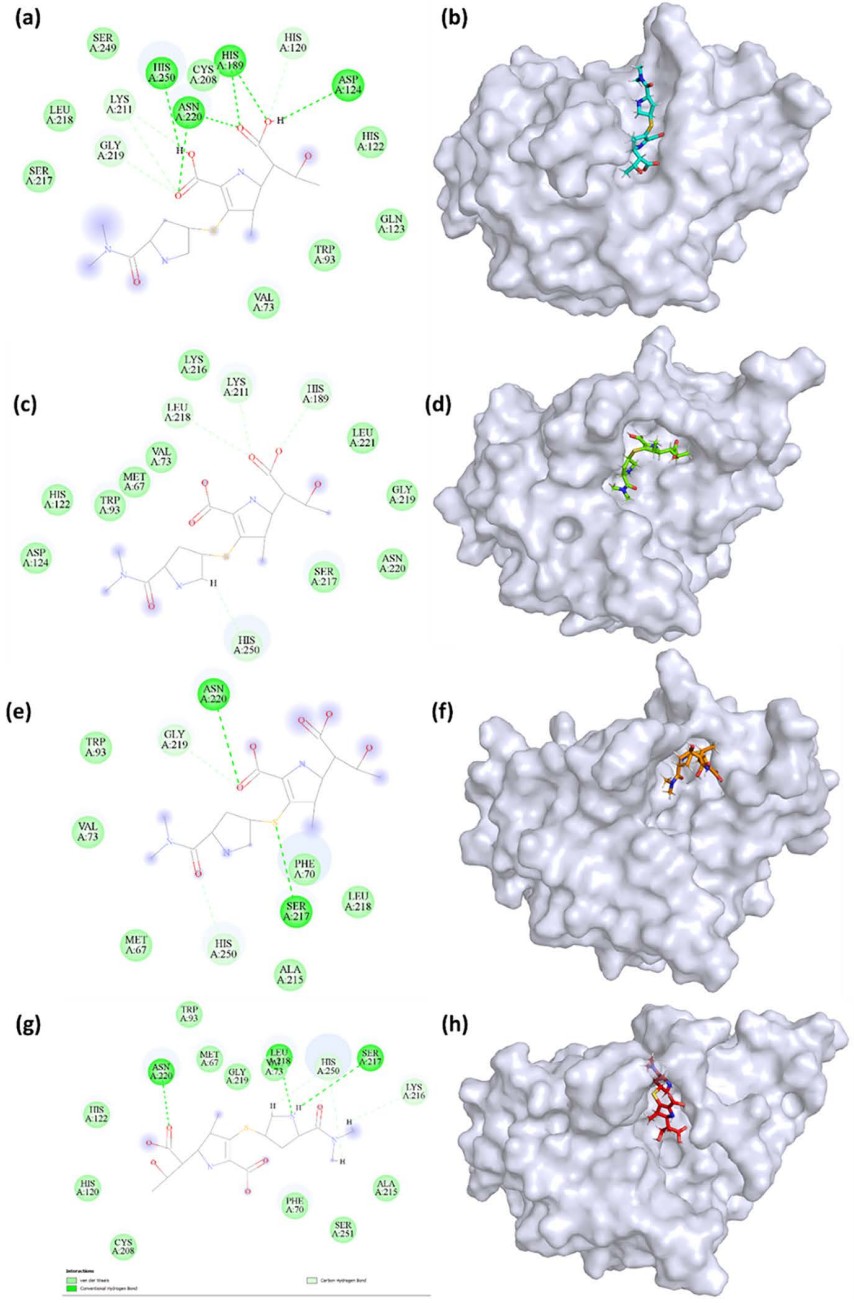

**Fig 7. 2D and 3D representation of the poses of (a, b) 0 ns pose (c, d) 100 ns pose (e, f) 200 ns pose (g, h) 300 ns pose of the control.**

meropenem suggests that this compound may act as a competitive inhibitor, possibly mimicking substrate-like binding patterns to occupy the catalytic pocket. Previous studies have identified key NDM-1 binding residues for various substrates and inhibitors, primarily using structure-based computational methods [51,52]. The NDM-1-nitrocefin complex interaction involved critical residues including Ile35, Trp93, His189, Cys208, Lys211, Gly219, Asn220, and His250. Similarly, the NDM-1-ampicillin complex involved Trp93, Asp124, Cys208, Gly219, Asn220, and His250, while the NDM-1-meropenem

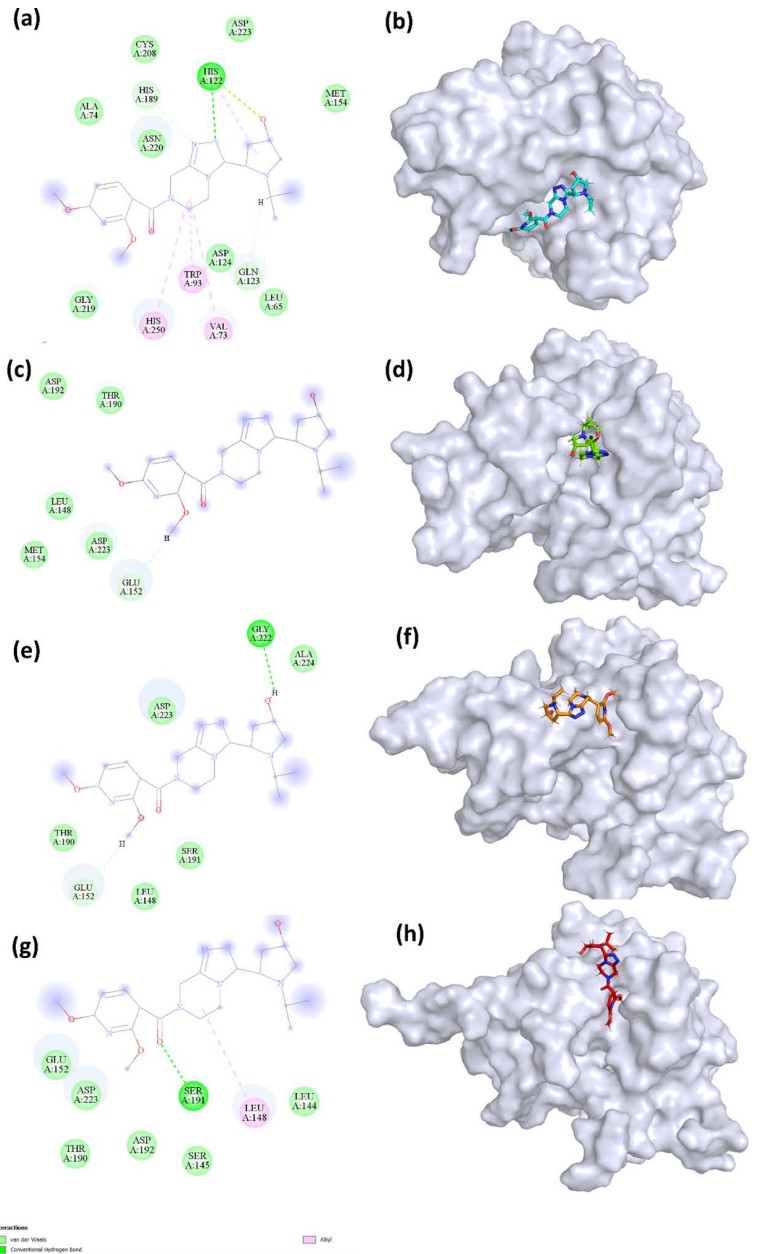

**Fig 8. 2D and 3D representation of the poses of (a, b) 0 ns pose (c, d) 100 ns pose (e, f) 200 ns pose (g, h) 300 ns pose of the S904-0022.**

complex was stabilized through interactions with Asp124, His120, Lys211, His122, His250, His189, Lys211, Val73, and Trp93. These antibiotics bind to the catalytic active site of NDM-1, leading to hydrolysis by the metallo-β-lactamase enzyme, which results in antibiotic resistance.

Several of these residues (Trp93, His250, His189, Val73) were previously identified as important for antibiotic binding and hydrolysis, indicating that **S904-0022** may be interacting in a key functional region of NDM-1. The recurrence of these residues across structurally diverse ligands highlights their conserved role in ligand recognition and enzymatic activity. Targeting such residues is a rational strategy in inhibitor design, as it leverages mechanistic insights from both substrate

and inhibitor binding. The common residue interactions of S904-0022 indicate that it binds in a functionally relevant region of NDM-1, interacting with key residues that are known to play a role in substrate recognition and enzyme function, making it a promising candidate for further inhibitor development. These findings suggest that further structural optimization of S904-0022, particularly focusing on enhancing interactions with His250, Asn220, and Trp93, could improve binding affinity and specificity, serving as a basis for future structure–activity relationship (SAR) studies.

**3.4.6. PCA.** PCA projection of the protein's conformational space during a 300 ns MDS is represented in Fig 9a. The plot displays the projections of the control and the ligand **S904-0022** on the first two principal components (eigenvectors). The x-axis corresponds to the projection on eigenvector 1, while the y-axis corresponds to the projection on eigenvector 2, both measured in nm. The protein-control complexes show tightly clustered in the conformational space, indicating minimal variation in conformation. The ligand **S904-0022** when bound to the protein results in a wider and more scattered distribution in the conformational space, suggesting an enhanced level of conformational diversity and flexibility. The PCA illustrates the protein's exploration of its structural space during the simulation. A higher degree of concentration suggests a narrower range of conformational alterations, whereas a more diffused distribution suggests a wider level of flexibility and exploration of other conformations. The wider dispersion of the blue dots observed in **S904-0022** indicates that this ligand elicits more significant alterations in the conformation and flexibility of the protein structure when compared to the control. It contributes to the understanding of protein-ligand interactions and their significance in protein function. The PCA projection demonstrates that the ligand **S904-0022** elicits more significant changes in the protein's structural flexibility compared to the control throughout the 300 ns molecular dynamics simulation.

**3.4.7. Free energy landscape.** The Fig 9b, c illustrates the free energy landscape (FEL) of the protein during a 300 nanosecond (ns) molecular dynamics (MD) simulation, comparing its behaviour in the presence of a control and the ligand **S904-0022**. Here, both scenarios are displayed for the first two main components (PC1 and PC2). The colour gradient corresponds to the free energy levels, where blue signifies lower energy states and red signifies higher energy states. The control's free energy landscape has several low-energy basins, represented by blue patches, which are localized in certain places. This suggests the presence of stable conformational states. The energy landscape exhibits a condensed structure, indicating a restricted range of conformational adaptability and the presence of stable protein states. Free energy landscape for **S904-0022** has a scattered pattern with multiple low-energy basins distributed over a larger region. The energy landscape is wider, suggesting more conformational flexibility and a greater variety of stable states in comparison to the control. The existence of clearly defined low-energy basins in the free energy landscape signifies areas of conformational stability where the protein tends to remain during the molecular dynamics simulation. The increased breadth and dispersion of the free energy landscape for **S904-0022** indicates that this ligand causes a greater degree of conformational diversity in the protein, resulting in a wider spectrum of stable structural states. This may suggest enhanced adaptability and possible alterations in protein functionality.

**3.4.8. Binding free energy.** The Fig 10 demonstrates the energy constituents of protein-ligand interactions. Fig 10a showed the energy components of the control when it was attached to the protein. The van der Waals interactions (VDWAALS) made a substantial contribution to the binding energy, with a value of −30.58 kcal/mol. The electrostatic interactions (EEL) contribute a value of −19.40 kcal/mol. The polar solvation energy (EGB) has a contribution of 35.46 kcal/mol, which acts in opposition to the binding process. The non-polar solvation energy (ESURF) has a contribution of −4.39 kcal/mol. The gas-phase energy (GGAS) has a significant contribution of −49.98 kcal/mol. The solvation energy (GSOLV) counteracts the binding with a magnitude of 31.08 kcal/mol. The control exhibits a net binding energy of −18.90 kcal/mol. Fig 10b displays the energy components in the presence of **S904-0022** when the protein is bound. The van der Waals interactions (VDWAALS) contribute −42.38 kcal/mol. The electrostatic interactions (EEL) exhibit a substantial value of −71.03 kcal/mol. The polar solvation energy (EGB) has a contribution of 83.29 kcal/mol, which acts in opposition to the binding process. The non-polar solvation energy (ESURF) was calculated to be −5.64 kcal/mol. The gas-phase energy (GGAS) has a significant contribution of −113.42 kcal/mol. The solvation energy (GSOLV) was responsible for a contribution of 77.65 kcal/mol. The total binding free energy is −35.77 kcal/mol.

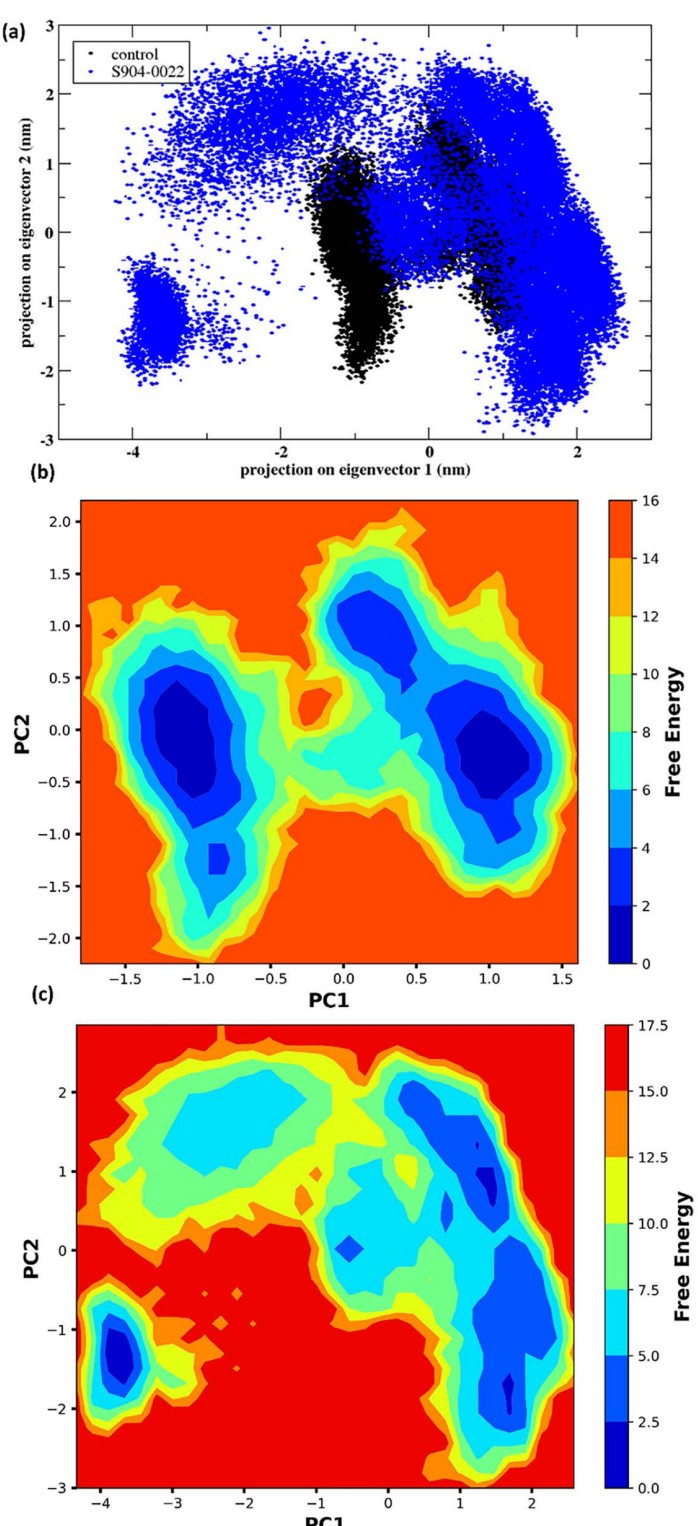

**Fig 9. (a) PCA of the protein-ligand complex and the control during the 300 ns simulation. The free energy landscape of the protein bound to the ligands and the control over the course of the 300 ns simulation is presented as follows: (b) Control, (c) S904-00220022.**

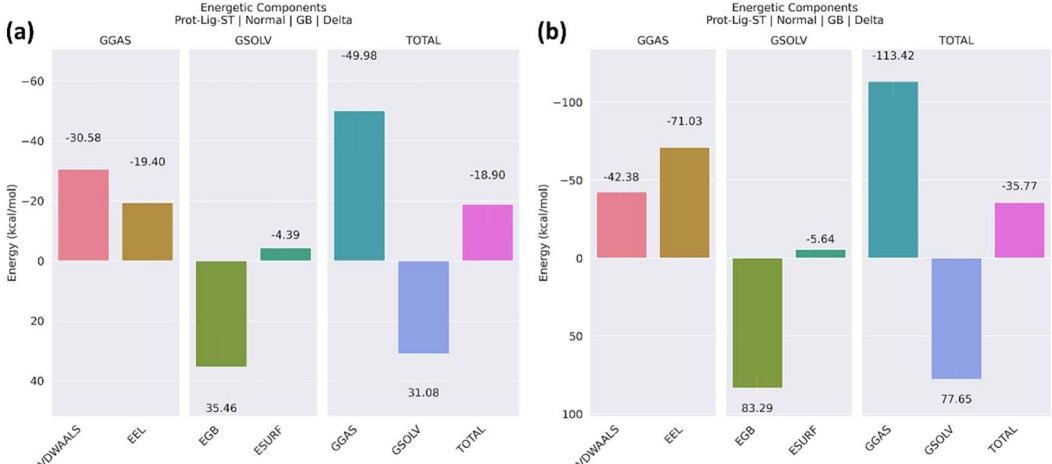

**Fig 10. The Binding-energy generated from the complexes of NDM-1 protein and ligands (a) Control, (b) S904-0022.**

S4 Fig presents per-residue energy decomposition for the control ligand and S904-0022 when bound to Aurora Kinase A, highlighting key differences in binding interactions. The binding energy contributions from individual residues provide insight into the stability and interaction strength of each ligand within the binding pocket. For the control ligand (S4(a) Fig), the strongest binding contribution comes from residues, such as Val73 (−1.32 kcal/mol), Asp124 (−1.68 kcal/mol), Leu218 (−1.61 kcal/mol), and Asn220 (−0.80 kcal/mol), contribute moderate stabilizing effects. However, Asp124 shows a positive energy contribution (+0.71 kcal/mol), suggesting a slightly unfavorable interaction or local repulsion. In contrast, **S904-0022** (S4(b) Fig) demonstrated a distinct interaction pattern, with ASP-223 (−3.87 kcal/mol) providing the strongest stabilizing effect, significantly greater than any interaction observed for the control ligand. Other contributing residues included Met154 (−0.28 kcal/mol), Asp124 (−0.04 kcal/mol), and Gln123 (−0.07 kcal/mol), suggesting additional stabilizing interactions, whereas His189 (0.10 kcal/mol) displayed a negligible destabilizing effect. The dominant stabilization from Asp223 in **S904-0022**, compared to the more distributed and moderate contributions in the control ligand, suggests that **S904-0022** binds in a distinct conformation that enhances its stability within the active site.

The Fig 10 illustrates the individual contributions of several energetic components to the total binding energy of the protein-ligand complexes. Negative values contribute favorably to binding, while positive values oppose binding. Comparing the identified ligands with the control, it was observed that **S904-0022** showed more favorable binding free energy than the control. The binding free energy of **S904-0022** was −35.77 kcal/mol while for the control it was −18.90 kcal/mol.

## 4. Limitations

This study offers a thorough computational methodology for identifying possible NDM-1 inhibitors; however, several limitations must be recognized. The predictions derived from QSAR modeling and molecular docking rely on theoretical assumptions and may not entirely encompass the intricacies of biological systems. Moreover, the predictions of binding affinity depend on scoring functions that may neglect significant entropic and solvation influences. Molecular dynamics simulations offer enhanced understanding of ligand–protein interactions; yet, they remain limited by timeframe and the precision of force fields. Future research will rectify these shortcomings by integrating a broader range of experimentally validated information into the QSAR model. Primarily, *in vitro* and *in vivo* validation will be emphasized to ascertain the biological significance of the projected hits and facilitate the conversion of computational leads into viable therapeutic candidates.

## 5. Future directions

In order to verify the computational findings of this study, we intend to implement *in vitro* biochemical assays, such as enzyme inhibition assays, to quantify the ability of **S904-0022** and other promising compounds to inhibit NDM-1 activity. In order to quantify β-lactamase hydrolysis and ascertain $IC_{50}$ values, we will employ a nitrocefin-based spectrophotometric assay. Furthermore, we plan to evaluate the antibacterial efficacy of our lead compounds by conducting minimum inhibitory concentration (MIC) assays against bacterial strains that produce NDM-1. We will conduct isothermal titration calorimetry (ITC) and surface plasmon resonance (SPR) to directly measure the binding affinity and thermodynamics of ligand-protein interactions in order to further confirm the binding interactions predicted by molecular docking and molecular dynamics simulations. Furthermore, in order to assess its potential as a therapeutic candidate, we will compare the inhibitory potency of **S904-0022** with previously reported NDM-1 inhibitors and clinically relevant β-lactamase inhibitors. Structural optimization and medicinal chemistry strategies will be implemented to optimize potency and selectivity in the event that these preliminary investigations indicate promising activity. The feasibility of **S904-0022** as a main compound for further preclinical and clinical development will be determined by the outcome of these experimental validations.

## 6. Conclusion

The limited treatment options available contribute to the high morbidity and mortality rates associated with the spread of NDM-1. Developing alternative therapeutic strategies targeting NDM-1-producing bacteria could significantly improve clinical outcomes by inhibiting NDM-1 with novel small molecules. This study aimed to identify natural product-derived inhibitors that effectively impede NDM-1 activity. An *in-silico* pipeline was designed to screen and prioritize compounds with high binding affinity and stability. Three lead candidates with strong predicted inhibitory potential against NDM-1 were identified from a natural product library. Among them, **S904-0022** demonstrated the most stable binding interactions during molecular dynamics simulations, forming key contacts with catalytically relevant residues. Additionally, **S904-0022** exhibited a favorable binding free energy profile and maintained conformational integrity within the active site. These properties, along with its structural drug-like characteristics, highlight **S904-0022** as a promising scaffold for further medicinal chemistry optimization and experimental validation in the search for potent NDM-1 inhibitors.

## Supporting information

**S1 Table. Compounds library used in this study.**
(XLSX)

**S2 Table. The training-test dataset.**
(XLSX)

**S3 Table. QSAR results.**
(XLSX)

**S4 Table. Top binding energies and normalized binding energies of the screened selected compounds along with control.**
(DOCX)

**S1 Fig. Validation of docking protocol by re-docking the co-crystallized ligand (ORV) into the active site of NDM-1 (PDB ID: 4EYL).** Legend: The protein backbone is shown in magenta cartoon representation. The original ligand conformation from the crystal structure is shown in green, while the re-docked pose is displayed in yellow.
(PNG)

**S2 Fig. 3D representation of the poses of (a) 0 ns pose (b) 100 ns pose (c) 200 ns pose (d) 300 ns pose of the N118-0137.**
(PNG)

**S3 Fig. 3D representation of the poses of (a) 0 ns pose (b) 100 ns pose (c) 200 ns pose (d) 300 ns pose of the S721-1034.**
(PNG)

**S4 Fig. Per-residue decomposition of (a) Control (b) S904-0022 when bound to the aurora kinase A.**
(PNG)

**S5 Table. Top binding energies and normalized binding energies of the screened selected compounds along with control.**
(XLSX)

**S6 Table. Cluster 1.**
(XLSX)

**S7 Table. Cluster 2.**
(XLSX)

**S8 Table. Cluster 3.**
(XLSX)

## Acknowledgments

This research has been funded by Scientific Research Deanship at University of Hail Saudi Arabia through project number RCP-24 028.

## Author contributions

**Conceptualization:** Emira Noumi, Mejdi Snoussi, Hisham N. Altayb, Muhammad Afzal, Vincenzo De Feo.

**Formal analysis:** Nouha Bouali, Mamdouh M. Alshammari, Hisham N. Altayb.

**Methodology:** Emira Noumi, Hisham N. Altayb, Muhammad Afzal.

**Project administration:** Emira Noumi, Hisham N. Altayb, Vincenzo De Feo.

**Software:** Hisham N. Altayb, Muhammad Afzal.

**Supervision:** Vincenzo De Feo.

**Validation:** Mejdi Snoussi, Nouha Bouali, Mamdouh M. Alshammari, Vincenzo De Feo.

**Visualization:** Mamdouh M. Alshammari.

**Writing – original draft:** Emira Noumi, Mejdi Snoussi, Nouha Bouali, Hisham N. Altayb, Muhammad Afzal.

**Writing – review & editing:** Emira Noumi, Mejdi Snoussi, Hisham N. Altayb, Muhammad Afzal.

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
