## [Decision Letter · Decision Letter 0]

PONE-D-25-11337Pharmacophore-based virtual screening, Docking, and MD simulation studies: An in-silico perspective for the identification of potential MBL inhibitorsPLOS ONE

Dear Dr. Snoussi,

Thank you for submitting your manuscript to PLOS ONE. After careful consideration, we feel that it has merit but does not fully meet PLOS ONE’s publication criteria as it currently stands. Therefore, we invite you to submit a revised version of the manuscript that addresses the points raised during the review process.

**ACADEMIC EDITOR: **In some sections of the manuscript including the title, the author mentioned "Pharmacophore-based virtual screening", what does this mean? Given that in the main texts of the objectives, methodology, results and discussion, nothing can be referenced to pharmacophore based screening or virtual screening by pharmacophore model, I recommend that this is expunged from the submission to avoid confusion. In addition, how was the docking study validated? All concerns raised by the reviewers need to be adequately addressed, after then, the manuscript can be reconsidered. 

We look forward to receiving your revised manuscript.

Kind regards,

Yusuf Oloruntoyin Ayipo, Ph.D

Academic Editor

PLOS ONE

“This research has been funded by Scientific Research Deanship at University of Hail Saudi Arabia through project number RCP-24 028.”

Additional Editor Comments:

In some sections of the manuscript including the title, the author mentioned "Pharmacophore-based virtual screening", what does this mean? Given that in the main texts of objectives, methodology, results and discussion, nothing can be referenced to pharmacophore based screening, I recommend that this is expunged from the submission to avoid confusion. In addition, how was the docking study validated? All concerns raised by the reviewers need to be adequately addressed, after that, the manuscript can be reconsidered.

Reviewers' comments:

Reviewer's Responses to Questions

**Comments to the Author**

1. Is the manuscript technically sound, and do the data support the conclusions?

Reviewer #1: Yes

Reviewer #2: Yes

Reviewer #3: Yes

Reviewer #4: Partly

2. Has the statistical analysis been performed appropriately and rigorously? 

Reviewer #1: Yes

Reviewer #2: Yes

Reviewer #3: Yes

Reviewer #4: Yes

3. Have the authors made all data underlying the findings in their manuscript fully available?

Reviewer #1: Yes

Reviewer #2: Yes

Reviewer #3: Yes

Reviewer #4: Yes

4. Is the manuscript presented in an intelligible fashion and written in standard English?

Reviewer #1: Yes

Reviewer #2: Yes

Reviewer #3: Yes

Reviewer #4: Yes

5. Review Comments to the Author

Reviewer #1: SECTION 1: PLOS ONE REVIEWER GUIDELINES

• What are the main claims of the paper and how significant are they for the discipline?

I think they are moderately significant.

• Are the claims properly placed in the context of the previous literature? Have the authors treated the literature fairly?

Yes, fairly well.

• Do the data and analyses fully support the claims? If not, what other evidence is required?

Yes, data supports claim.

• PLOS ONE encourages authors to publish detailed protocols and algorithms as supporting information online. Do any particular methods used in the manuscript warrant such treatment? If a protocol is already provided, for example for a randomized controlled trial, are there any important deviations from it? If so, have the authors explained adequately why the deviations occurred?

Adequately, but more information / request is provided in notes below.

• If the paper is considered unsuitable for publication in its present form, does the study itself show sufficient potential that the authors should be encouraged to resubmit a revised version?

Yes, the paper is good for publication upon minor revision detailed below.

• Are original data deposited in appropriate repositories and accession/version numbers provided for genes, proteins, mutants, diseases, etc.?

Original data are cited. More requests in notes below.

• Does the study conform to any relevant guidelines such as CONSORT, MIAME, QUORUM, STROBE, and the Fort Lauderdale agreement?

Does not seem applicable here.

• Are details of the methodology sufficient to allow the experiments to be reproduced?

Nearly so; suggestions for improvement provided in notes below.

• Is any software created by the authors freely available?

Does not seem applicable here.

• Is the manuscript well organized and written clearly enough to be accessible to non-specialists?

Nearly so; suggestions for improvement provided in notes below.

• Is it your opinion that this manuscript contains an NIH-defined experiment of Dual Use concern?

Not at this time.

SECTION 2: MY COMMENTS / RECOMMENDATIONS / REQUESTS

• For the sake of readability, I ask that the Authors rephrase the first sentence of the abstract to clarify that antibiotic resistance is what is being proliferated (if that is the intent of the Authors), and not that the very identification of NDM-1 causes a proliferation of antibiotic resistance. Also, I ask that all abbreviations are defined throughout the manuscript, including in the abstract (examples include ML, CHARMM36, CGneFF, LINCS, NVT, NPT, PC, and PCA).

• As with all abbreviations, I recommend writing techniques like “quantitative structure-activity relationship” (QSAR) and “molecular dynamics” (MD) in lower case.

• Line 67 – “Expanding upon this basis, a recent work employed a multi-step virtual screening method for the identification of non-β-lactam inhibitors against NDM-1.” Lacking citation to the “recent work”.

• Line 76-81 – “The objective of this investigation was to employ computational methods to comprehensively investigate and identify natural product compounds that exhibit inhibitory activity against NDM-1.” I recommend rewording as “identify and investigate”, in consistence with the flow of the story. Also, I recommend caution on the use of past tense to describe the objective of the study. Finally, there seem to be a duplicate statement of objective, I therefore recommend making that section more succinct and thus avoid potential confusion.

• Line 83 – I recommend Authors should not make the claim of “strong interactions” until the results have been presented (and discussed).

• Line 84 – For clarity and readability, and to avoid bogus claims, I recommend the language of “identified ‘an inhibitor’”, not ‘the inhibitor’

• Line 141 – I recommend that Authors should refrain from hyphenating or abbreviating molecular dynamics simulations; it is not conventional to do so. Same issue in line 169 with free energy landscape.

• Line 149 – For readability, Authors should consider rewording “for the neutralization its charge”

• Line 152 – The “raised to” statement did not initially clarify a reference temperature. I strongly recommend that Authors make clear and simple the procedural steps involved in all analyses.

• Line 152 – “In addition, the entire system was raised to a temperature of 310 K using a timestep of 2 fs for a simulation time of 100 ps in the NVT ensemble and pressure (NPT) for a duration of 1 ns each at 310 K and 1 atmosphere.”

The statement seem to lack clarity and contains too many contradictory contents; some thoughts: what is the reference temperature of the post-minimization simulation, was there an annealing step, how long were the isothermal or annealing steps (what is the distinction between the 100 ps and 1 ns mentioned), what was the timestep for each? These were not immediately clear upon first-time reading, hence the need to detail all procedures in plain language as much as possible.

• Line 155 – Authors should use conventional language; “production” rather than “manufacturing” for the data collection phase of the molecular dynamics simulations

• Line 156 – To enhance clarity and readability, Authors should consider rewording “Velocity scaling employed to ensure a constant environment of simulation”. The statement should also be clarified to be pertaining to system temperature and not merely ‘environment of simulation’.

• Line 159 – “The hydrogen bonding patterns within the protein-ligand complex were examined utilizing GROMACS' internal tools in order to gain a more comprehensive understanding of the dynamic interactions” – Authors should mention the specific GROMACS tool being used here, and describe its implementation. Also, Authors should stay consistent in representing the GROMACS software in uppercase—no need to provide full meaning (line 142).

• Line 176 – To enhance readability, Authors should consider placing a comma after “kB”

• Line 197 – To enhance readability, Authors should consider replacing “The that” with “The”

• Line 210 – Authors should consider supplying the appropriate citation to PyMol (and to every software being utilized for the study—as was done for GROMACS)

• Authors should consider including the unit of the distance column in Table 2.

• Section 3.4: RMSD – To strengthen claim, I recommend that the Authors make a short, informational comment (with citation(s)) on the potential effects of the protein instability that has been induced by ligand interactions, thus placing that protein stability information in proper context.

• I recommend that the Authors include the molecular structures of all four ligands (the control as well as the three identified) somewhere appropriate in the manuscript main. Also, was the control molecule the same for all analysis performed?

• Line 409 – Could the Authors make comments or provide citations to justify that “1-2 hydrogen bonds” is sufficient for S904-0022 to be deemed relevant for further analyses including in-vitro.

Reviewer #2: The manuscript presents a comprehensive computational study aimed at identifying potential inhibitors of New Delhi Metallo-beta-lactamase-1 (NDM-1) using pharmacophore-based virtual screening, molecular docking, and molecular dynamics (MD) simulations. The study is technically sound, with robust methodologies and data supporting the conclusions. Below is a summary of the review comments and recommendations:

STRENGHTS:

Methodology: The study employs a well-structured, multi-step computational approach, including machine learning-based QSAR modeling, molecular docking, and MD simulations.

Data-Driven Conclusions: The conclusions are supported by extensive data analysis, including binding free energy calculations and interaction analysis.

Validation and Controls: The use of a control molecule (ORV) and cross-validation for the QSAR model enhances the validity of the findings.

Clarity and Structure: The manuscript is well-organized and generally well-written, with clear sections and effective use of figures and tables.

AREAS OF IMPROVEMENTS:

Statistical Analysis:

Provide additional validation metrics for the QSAR model (e.g., RMSE, MAE).

Include confidence intervals or error estimates for key metrics.

Consider performing statistical tests to compare binding affinities and stability.

Data Availability:

Ensure all raw data (e.g., docking scores, MD trajectories, QSAR model data) are included in the supplementary materials or deposited in a public repository.

Clearly state any restrictions on data sharing in the Data Availability Statement.

Language and Clarity:

Correct minor grammatical errors and improve sentence structure.

Simplify overly complex sentences and avoid repetition.

Ensure consistency in terminology (e.g., use "beta" or "β" consistently).

Figures and Tables:

Improve clarity of labels and legends in some figures (e.g., Figure 3).

Future Directions:

Briefly discuss potential limitations of the computational approach and how these will be addressed in future work.

Reviewer #3: The topic is highly relevant given the global public health threat posed by antibiotic resistance, particularly from NDM-1-producing bacteria. The study's use of advanced computational tools to address this challenge is commendable. However, there are several areas where clarity, scientific rigor, and presentation can be improved. These include methodological details, justification of certain choices, and deeper discussion of results in the context of prior literature. Additionally, minor language and formatting issues detract from readability and should be addressed.

For an instance, the authors need to provide justifications for some of the parameters and methods used in the analysis. e.g docking parameters, MACCS keys, k=3 in k-means clustering, etc.

Reviewer #4: The manuscript presents a computational approach for identifying potential NDM-1 inhibitors, leveraging machine learning-based QSAR modeling, molecular docking, and MD simulations. The study is well-structured and relevant, but certain areas need improvement. The QSAR model should provide more details on feature selection, data preprocessing, and validation metrics such as RMSE or MAE to enhance reproducibility.

Additionally, molecular docking parameters, including ligand flexibility, scoring functions, and docking poses, should be clearly specified. While the results suggest strong binding affinities, binding free energy calculations would be more robust if confidence intervals or standard deviations were included.

Furthermore, benchmarking against known NDM-1 inhibitors is necessary to assess the novelty and effectiveness of the proposed compounds. Some figures, such as RMSD and FEL plots, need clearer axis labeling and statistical annotations for better interpretation.

The manuscript is generally well-written, though minor grammatical issues and abstract restructuring could improve readability. It has strong potential but requires major revisions to enhance computational clarity, statistical validation, and data presentation before publication.

6. PLOS authors have the option to publish the peer review history of their article (what does this mean? ). If published, this will include your full peer review and any attached files.

**Do you want your identity to be public for this peer review?** For information about this choice, including consent withdrawal, please see our Privacy Policy .

Reviewer #1: No

Reviewer #2: No

Reviewer #3: No

Reviewer #4: **Yes: ** Abdulquadir Aderinto

---

## [Author Response · Author response to Decision Letter 1]

12 Apr 2025

Reviewer #1:

SECTION 2: MY COMMENTS / RECOMMENDATIONS / REQUESTS

Comment 1. For the sake of readability, I ask that the Authors rephrase the first sentence of the abstract to clarify that antibiotic resistance is what is being proliferated (if that is the intent of the Authors), and not that the very identification of NDM-1 causes a proliferation of antibiotic resistance. Also, I ask that all abbreviations are defined throughout the manuscript, including in the abstract (examples include ML, CHARMM36, CGneFF, LINCS, NVT, NPT, PC, and PCA).

Response: We appreciate the reviewer’s suggestion. The first sentence of the abstract has been rephrased to clarify that it is the spread of antibiotic-resistant infections, not the identification of NDM-1 itself, that is of concern. Additionally, all abbreviations mentioned—ML, CHARMM36, CGenFF, LINCS, NVT, NPT, PC, and PCA—have been defined at their first appearance in both the abstract and main text to enhance clarity and readability.

Comment 2. As with all abbreviations, I recommend writing techniques like “quantitative structure-activity relationship” (QSAR) and “molecular dynamics” (MD) in lower case.

Response: As recommended by the reviewer, the techniques were written in lower case.

Comment 3. Line 67 – “Expanding upon this basis, a recent work employed a multi-step virtual screening method for the identification of non-β-lactam inhibitors against NDM-1.” Lacking citation to the “recent work”.

Response: The citations have been added. Kindly review the revised manuscript.

Comment 4. Line 76-81 – “The objective of this investigation was to employ computational methods to comprehensively investigate and identify natural product compounds that exhibit inhibitory activity against NDM-1.” I recommend rewording as “identify and investigate”, in consistence with the flow of the story. Also, I recommend caution on the use of past tense to describe the objective of the study. Finally, there seem to be a duplicate statement of objective, I therefore recommend making that section more succinct and thus avoid potential confusion.

Response: The paragraph has been modified accordingly. Kindly review the revised manuscript.

Comment 5. Line 83 – I recommend Authors should not make the claim of “strong interactions” until the results have been presented (and discussed).

Response: The sentences have been modified accordingly. Kindly review the revised manuscript.

Comment 6. Line 84 – For clarity and readability, and to avoid bogus claims, I recommend the language of “identified ‘an inhibitor’”, not ‘the inhibitor’

Response: The sentences have been modified accordingly. Kindly review the revised manuscript.

Comment 7. Line 141 – I recommend that Authors should refrain from hyphenating or abbreviating molecular dynamics simulations; it is not conventional to do so. Same issue in line 169 with free energy landscape.

Response: The sentences have been modified accordingly. Kindly review the revised manuscript.

Comment 8. Line 149 – For readability, Authors should consider rewording “for the neutralization its charge”

Response: The sentences have been modified accordingly. Kindly review the revised manuscript.

Comment 9. Line 152 – The “raised to” statement did not initially clarify a reference temperature. I strongly recommend that Authors make clear and simple the procedural steps involved in all analyses.

Response: The sentences have been modified accordingly. Kindly review the revised manuscript.

Comment 10. Line 152 – “In addition, the entire system was raised to a temperature of 310 K using a timestep of 2 fs for a simulation time of 100 ps in the NVT ensemble and pressure (NPT) for a duration of 1 ns each at 310 K and 1 atmosphere.”

The statement seem to lack clarity and contains too many contradictory contents; some thoughts: what is the reference temperature of the post-minimization simulation, was there an annealing step, how long were the isothermal or annealing steps (what is the distinction between the 100 ps and 1 ns mentioned), what was the timestep for each? These were not immediately clear upon first-time reading, hence the need to detail all procedures in plain language as much as possible.

Response: The sentences have been modified accordingly to clarify the details of the procedure. Kindly review the revised manuscript.

Comment 11. Line 155 – Authors should use conventional language; “production” rather than “manufacturing” for the data collection phase of the molecular dynamics simulations

Response: The sentences have been modified accordingly. Kindly review the revised manuscript.

Comment 12. Line 156 – To enhance clarity and readability, Authors should consider rewording “Velocity scaling employed to ensure a constant environment of simulation”. The statement should also be clarified to be pertaining to system temperature and not merely ‘environment of simulation’.

Response: The sentences have been modified accordingly. Kindly review the revised manuscript.

Comment 13. Line 159 – “The hydrogen bonding patterns within the protein-ligand complex were examined utilizing GROMACS' internal tools in order to gain a more comprehensive understanding of the dynamic interactions” – Authors should mention the specific GROMACS tool being used here, and describe its implementation. Also, Authors should stay consistent in representing the GROMACS software in uppercase—no need to provide full meaning (line 142).

Response: The sentences have been modified accordingly. Kindly review the revised manuscript.

Comment 14. Line 176 – To enhance readability, Authors should consider placing a comma after “kB”

Response: The sentences have been modified accordingly. Kindly review the revised manuscript.

Comment 15. Line 197 – To enhance readability, Authors should consider replacing “The that” with “The”

Response: The sentences have been modified accordingly. Kindly review the revised manuscript.

Comment 16. Line 210 – Authors should consider supplying the appropriate citation to PyMol (and to every software being utilized for the study—as was done for GROMACS)

Response: The citations have been added accordingly. Kindly review the revised manuscript.

Comment 17. Authors should consider including the unit of the distance column in Table 2.

Response: The sentences have been modified accordingly. Kindly review the revised manuscript.

Comment 18. Section 3.4: RMSD – To strengthen claim, I recommend that the Authors make a short, informational comment (with citation(s)) on the potential effects of the protein instability that has been induced by ligand interactions, thus placing that protein stability information in proper context.

Response: The sentences have been added accordingly with citation. Kindly review the revised manuscript.

Comment 19. I recommend that the Authors include the molecular structures of all four ligands (the control as well as the three identified) somewhere appropriate in the manuscript main. Also, was the control molecule the same for all analysis performed?

Response: Thank you for the helpful suggestion. The molecular structures of the control compound (meropenem/ORV) and the three lead compounds (S904-0022, S721-1034, and N118-0137) have now been included in the main manuscript as a new figure. Additionally, we confirm that the same control molecule (ORV) was consistently used throughout all analyses, including molecular docking, molecular dynamics simulations, and binding free energy calculations. Kindly review the revised manuscript.

Comment 20. Line 409 – Could the Authors make comments or provide citations to justify that “1-2 hydrogen bonds” is sufficient for S904-0022 to be deemed relevant for further analyses including in-vitro.

Response: The sentences have been modified accordingly. Kindly review the revised manuscript.

Reviewer #2: The manuscript presents a comprehensive computational study aimed at identifying potential inhibitors of New Delhi Metallo-beta-lactamase-1 (NDM-1) using pharmacophore-based virtual screening, molecular docking, and molecular dynamics (MD) simulations. The study is technically sound, with robust methodologies and data supporting the conclusions. Below is a summary of the review comments and recommendations:

STRENGHTS:

Methodology: The study employs a well-structured, multi-step computational approach, including machine learning-based QSAR modeling, molecular docking, and MD simulations.

Data-Driven Conclusions: The conclusions are supported by extensive data analysis, including binding free energy calculations and interaction analysis.

Validation and Controls: The use of a control molecule (ORV) and cross-validation for the QSAR model enhances the validity of the findings.

Clarity and Structure: The manuscript is well-organized and generally well-written, with clear sections and effective use of figures and tables.

Response: Thank you for your positive and encouraging feedback. We sincerely appreciate the recognition of our methodological approach, data analysis, validation strategies, and overall clarity of the manuscript. Your comments are highly motivating and have helped us further refine the quality of our work.

AREAS OF IMPROVEMENTS:

Statistical Analysis:

Provide additional validation metrics for the QSAR model (e.g., RMSE, MAE).

Include confidence intervals or error estimates for key metrics.

Consider performing statistical tests to compare binding affinities and stability.

Response: As suggested by the reviewer, more details of the QSAR model has been provided. Kindly review the “2.2.ML-based QSAR Model” and “3.2.ML-based QSAR Model” of the revised manuscript.

We adopted a normalized scoring strategy to provide a relative and comparative assessment of binding affinities across all compounds. Kindly review “2.3. Virtual Screening” and “3.3. Molecular Docking” of the revised manuscript.

Data Availability:

Ensure all raw data (e.g., docking scores, MD trajectories, QSAR model data) are included in the supplementary materials or deposited in a public repository.

Clearly state any restrictions on data sharing in the Data Availability Statement.

Response: We appreciate the reviewer’s suggestion regarding data transparency. All relevant raw data, including docking scores, QSAR model outputs, and key molecular dynamics (MD) trajectory files, have been provided in the Supplementary Materials. Additionally, larger datasets and trajectory files have been deposited in a publicly accessible repository [ ]. We have also included a Data Availability Statement in the manuscript, which outlines access details and confirms that there are no restrictions on data sharing. Kindly review the revised manuscript.

Language and Clarity:

Correct minor grammatical errors and improve sentence structure.

Simplify overly complex sentences and avoid repetition.

Ensure consistency in terminology (e.g., use "beta" or "β" consistently).

Response: The terminology was made consistent, sentences were corrected and repetition were simplified. Kindly review the revised manuscript.

Figures and Tables:

Improve clarity of labels and legends in some figures (e.g., Figure 3).

Response: The labels and legends of Figures were improved. Kindly review the revised manuscript.

Future Directions:

Briefly discuss potential limitations of the computational approach and how these will be addressed in future work.

Response: The potential limitations and the future work to address it has been added. Kindly review the “4. Limitations” and “5. Future Directions” sections of the revised manuscript.

Reviewer #3: The topic is highly relevant given the global public health threat posed by antibiotic resistance, particularly from NDM-1-producing bacteria. The study's use of advanced computational tools to address this challenge is commendable. However, there are several areas where clarity, scientific rigor, and presentation can be improved. These include methodological details, justification of certain choices, and deeper discussion of results in the context of prior literature. Additionally, minor language and formatting issues detract from readability and should be addressed. For an instance, the authors need to provide justifications for some of the parameters and methods used in the analysis. e.g docking parameters, MACCS keys, k=3 in k-means clustering, etc.

Response: We sincerely thank the reviewer for acknowledging the relevance of our study and the application of advanced computational tools to address antibiotic resistance. In response to the concerns raised, we have made substantial revisions throughout the manuscript to improve clarity, scientific rigor, and presentation. Kindly review the revised manuscript.

Deeper discussion of prior literature was added. Kindly review the lines starting with “Numerous prior studies have examined…” and “A previous investigation revealed that ZINC05683641…” of the revised manuscript.

The justification for choosing docking parameters, MACCS keys, k=3 in k-means clustering were mentioned in the methodology section. Kindly review the “2.2.ML-based QSAR Model”, “ 2.4. Tanimoto Similarity and Clustering” and “2.3. Virtual Screening” of the revised manuscript.

Reviewer #4: The manuscript presents a computational approach for identifying potential NDM-1 inhibitors, leveraging machine learning-based QSAR modeling, molecular docking, and MD simulations. The study is well-structured and relevant, but certain areas need improvement.

Comment 1: The QSAR model should provide more details on feature selection, data preprocessing, and validation metrics such as RMSE or MAE to enhance reproducibility.

Response: As suggested by the reviewer, more details of the QSAR model has been provided. Kindly review the “2.2.ML-based QSAR Model” and “3.2.ML-based QSAR Model” of the revised manuscript.

Comment 2: Additionally, molecular docking parameters, including ligand flexibility, scoring functions, and docking poses, should be clearly specified. While the results suggest strong binding affinities, binding free energy calculations would be more robust if confidence intervals or standard deviations were included.

Response: The molecular docking parameters, including ligand flexibility, scoring functions, and docking poses were explained in details in the revised manuscript.

While standard deviations and confidence intervals would enhance the robustness of binding energy predictions, their calculation requires multiple independent docking runs per compound. For this study, such an approach performing repeated docking simulations for all 699 screened compounds along with the control ligand was not feasible due to computational resource limitations. Instead, we adopted a normalized scoring strategy to provide a relative and comparative assessment of binding affinities across all compounds.

Kindly review “2.3. Virtual Screening” and “3.3. Molecular Docking” of the revised manuscript.

Comment 3: Furthermore, benchmarking against known NDM-1 inhibitors is necessary to assess the novelty and effectiveness of the proposed compounds. Some figures, such as RMSD and FEL plots, need clearer axis labeling and statistical annotations for better interpretation.

Response: The Figures were improved. Kindly review the revised manuscript.

Comment 4: The manuscript is generally well-written, though minor grammatical issues and abstract restructuring could improve readability. It has strong potential but requires major revisions to enhance computational clarity, statistical validation, and data presentation before publication.

Response: The manuscript has been revised for grammar, abstract clarity, and improved presentation. Kindly review the “Abstract” of the revised manuscript.

Additional Editor Comments:

In some sections of the manuscript including the title, the author mentioned "Pharmacophore-based virtual screening", what does this mean? Given that in the main texts of objectives, methodology, results and discussion, nothing can be referenced to pharmacophore based screening, I recommend that this is expunged from the submission to avoid confusion. In addition, how was the docking study validated? All concerns raised by the reviewers need to be adequately addressed, after that, the manuscript can

---

## [Decision Letter · Decision Letter 1]

Structure-Based Virtual Screening, Molecular Docking, and MD Simulation Studies: An In-Silico Approach for Identifying Potential MBL Inhibitors

PONE-D-25-11337R1

Dear Dr. Snoussi,

We’re pleased to inform you that your manuscript has been judged scientifically suitable for publication and will be formally accepted for publication once it meets all outstanding technical requirements.

Kind regards,

Yusuf Oloruntoyin Ayipo, Ph.D

Academic Editor

PLOS ONE

Additional Editor Comments (optional):

The submission has scientific relevance as a potential starting point in search for new antibiotics to combat the MBL-inclined antibiotic resistance. The manuscript is well-prepared and the authors have responded positively to all concerns initially raised by the editor and respective reviewers. The quality has improved significantly up to the standard for publication in this title. I hereby recommend its acceptance for publication in the current form provided it passes other editorial checks.

Reviewers' comments:

Reviewer's Responses to Questions

**Comments to the Author**

1. If the authors have adequately addressed your comments raised in a previous round of review and you feel that this manuscript is now acceptable for publication, you may indicate that here to bypass the “Comments to the Author” section, enter your conflict of interest statement in the “Confidential to Editor” section, and submit your "Accept" recommendation.

Reviewer #3: All comments have been addressed

2. Is the manuscript technically sound, and do the data support the conclusions?

Reviewer #3: Yes

3. Has the statistical analysis been performed appropriately and rigorously? 

Reviewer #3: Yes

4. Have the authors made all data underlying the findings in their manuscript fully available?

Reviewer #3: Yes

5. Is the manuscript presented in an intelligible fashion and written in standard English?

Reviewer #3: Yes

6. Review Comments to the Author

Reviewer #3: The authors have addressed my concerns in the first review and I believe it should be accepted for a publication.

7. PLOS authors have the option to publish the peer review history of their article (what does this mean? ). If published, this will include your full peer review and any attached files.

**Do you want your identity to be public for this peer review?** For information about this choice, including consent withdrawal, please see our Privacy Policy .

Reviewer #3: No

---

## [Editor Report · Acceptance letter]

PONE-D-25-11337R1

PLOS ONE

Dear Dr. Snoussi,

I'm pleased to inform you that your manuscript has been deemed suitable for publication in PLOS ONE. Congratulations! Your manuscript is now being handed over to our production team.

Kind regards,

on behalf of

Dr. Yusuf Oloruntoyin Ayipo

Academic Editor

PLOS ONE